# Optimal Protocols for Continual Learning via Statistical Physics and Control Theory

**Francesco Mori**[1]**, Stefano Sarao Mannelli**[2,3]**, Francesca Mignacco**[4,5]

## Abstract

Artificial neural networks often struggle with *catastrophic forgetting* when learning multiple tasks sequentially, as training on new tasks degrades the performance on previously learned tasks. Recent theoretical work has addressed this issue by analysing learning curves in synthetic frameworks under predefined training protocols. However, these protocols relied on heuristics and lacked a solid theoretical foundation assessing their optimality. In this paper, we fill this gap by combining exact equations for training dynamics, derived using statistical physics techniques, with optimal control methods. We apply this approach to teacher-student models for continual learning and multi-task problems, obtaining a theory for task-selection protocols maximising performance while minimising forgetting. Our theoretical analysis offers non-trivial yet interpretable strategies for mitigating catastrophic forgetting, shedding light on how optimal learning protocols modulate established effects, such as the influence of task similarity on forgetting. Finally, we validate our theoretical findings with experiments on real-world data.

## 1 Introduction

Mastering a diverse range of problems is crucial for both artificial and biological systems. In the context of training a neural network on a series of tasks—a.k.a. *multi-task learning* (Caruana, 1993; 1994b;a; 1997)—the ability to learn new tasks can be improved by leveraging knowledge from previous ones (Suddarth & Kergosien, 1990). However, this process can lead to *catastrophic forgetting*, where learning new tasks degrades performance on older ones. This phenomenon has been observed in both theoretical neuroscience (McCloskey & Cohen, 1989; Ratcliff, 1990) and machine learning (Srivastava et al., 2013; Goodfellow et al., 2014), and occurs when the network parameters encoding older tasks are overwritten while training on a new task. Several mitigation strategies have been proposed (French, 1999; Kemker et al., 2018), including semi-distributed representations (French, 1991; 1992), regularisation methods (Kirkpatrick et al., 2017; Zenke et al., 2017; Li & Hoiem, 2017), dynamical architectures (Zhou et al., 2012; Rusu et al., 2016), and others (see e.g. Parisi et al. (2019); De Lange et al. (2021) for thorough reviews). A common strategy, known as *replay*, is to present the network with examples from the old tasks while training on the new one to minimise forgetting (Shin et al., 2017; Draelos et al., 2017; Rolnick et al., 2019).

On the theoretical side, Baxter (2000) pioneered the research on continual learning by deriving PAC bounds. More recently, further performance bounds have been obtained in the context of multi-task learning, few-shot learning, domain adaptation, and hypothesis transfer learning (Wang et al., 2020; Zhang & Yang, 2021; Zhang & Gao, 2022). However, these results focused on worst-case analysis, offering bounds that may not reflect the typical performance of algorithms. In contrast, Dhifallah & Lu (2021) began investigating the typical-case scenario, providing a precise characterisation of transfer learning in simple neural network models. Gerace et al. (2022); Ingrosso et al. (2024) extended this analysis to more complex architectures and generative models, allowing for a better description of the relation between tasks. Finally, Lee et al. (2021; 2022) proposed a theoretical

[1]Rudolf Peierls Centre for Theoretical Physics, University of Oxford, Oxford OX1 3PU, United Kingdom
[2]Data Science and AI, Computer Science and Engineering, Chalmers University of Technology and University of Gothenburg
[3]School of Computer Science and Applied Mathematics, University of the Witwatersrand
[4]Graduate Center, City University of New York, New York, NY 10016, USA
[5]Joseph Henry Laboratories of Physics, Princeton University, Princeton, NJ 08544, USA

framework for the study of the dynamics of continual learning with a focus on catastrophic forgetting. Their work provided a theoretical explanation for the surprising empirical results of Ramasesh et al. (2020), which revealed a non-monotonic relation between forgetting and task similarity, where maximal forgetting occurs at intermediate task similarity. Analogously, Shan et al. (2024) studied a Gibbs formulation of continual learning in deep linear networks, and demonstrated how the interplay between task similarity and network architecture influences forgetting and knowledge transfer.

Despite the significant interest in transfer learning and catastrophic forgetting, mitigation strategies considered thus far were pre-defined heuristics, offering no guarantees of optimality. In contrast, here we address the problem of identifying the optimal protocol to minimise forgetting. Specifically, we focus on replay as a prototypical mitigation strategy. We use optimal control theory to determine the optimal training protocol that maximises performance across different tasks.

**Our contribution.** In this work, we combine dimensionality-reduction techniques from statistical physics (Saad & Solla, 1995a;b; Biehl & Schwarze, 1995) and Pontryagin's maximum principle from control theory (Feldbaum, 1955; Pontryagin, 1957; Kopp, 1962). This approach allows us to derive optimal task-selection protocols for the training dynamics of a neural network in a continual learning setting. Pontryagin's principle works efficiently in low-dimensional deterministic systems. Hence, applying it to neural networks requires the statistical physics approach (Engel, 2001), which reduces the evolution of high-dimensional stochastic systems to a few key order parameters governed by ordinary differential equations (ODEs) (Saad & Solla, 1995a;b; Biehl & Schwarze, 1995). Specifically, we consider the teacher-student framework of Lee et al. (2021)—a prototype continual learning setting amenable to analytic characterisation. Our main contributions are:

- We leverage the ODEs for the learning curves of online SGD to derive closed-form formulae for the optimal training protocols. In particular, we provide equations for the optimal task-selection protocol and the optimal learning rate schedule, as a function of the task similarity $\gamma$ and the problem parameters. Our framework is broadly applicable beyond the specific context of continual learning, and we outline several potential extensions.

- We evaluate our equations for a range of problem parameters and find highly structured protocols. Interestingly, we are nonetheless able to interpret these strategies a posteriori, formulating a criterion for "pseudo-optimal" task-selection. This strategy consists of an initial *focus* phase, where only the new task is presented to the network, followed by a *revision* phase, where old tasks are replayed.

- We clarify the impact of task similarity on catastrophic forgetting. At variance with what previously observed (Ramasesh et al., 2020; Lee et al., 2021; 2022), catastrophic forgetting is minimal at intermediate task similarity when learning with optimal task selection. We provide a mechanistic explanation of this phenomenon by disentangling dynamical effects at the level of first-layer and readout weights.

- We demonstrate that the insights from our optimal strategies in synthetic settings transfer to real datasets. Specifically, we show the efficacy of our pseudo-optimal strategy on a continual learning task using the Fashion-MNIST dataset. Here, the pseudo-optimal strategy effectively interpolates between simple heuristics depending on the problem's parameters.

**Further related works.** Recent theoretical works on online dynamics in one-hidden-layer neural networks have addressed various learning problems, including over-parameterisation (Goldt et al., 2019), algorithmic analysis (Refinetti et al., 2021; Srinivasan et al., 2024), and learning strategies (Lee et al., 2021; Saglietti et al., 2022; Sarao Mannelli et al., 2024). However, these studies have not explored the problem from an optimal control perspective.

Early works addressed the optimality of hyperparameters in high-dimensional online learning for committee machines via control theory. These studies focused on optimising the learning rate (Saad & Rattray, 1997; Rattray & Saad, 1998; Schlösser et al., 1999), the regularisation (Saad & Rattray, 1998), and the learning rule (Rattray & Saad, 1997). However, to the best of our knowledge, the problem of optimal task selection has not been explored yet. Carrasco-Davis et al. (2023) and Li et al. (2024) applied optimal control to the dynamics of connectionist models of behaviour, but their analysis was limited to low-dimensional and finite-dimensional settings. Urbani (2021) extended the Bellman equation to high-dimensional mean-field dynamical systems, though without considering learning processes.

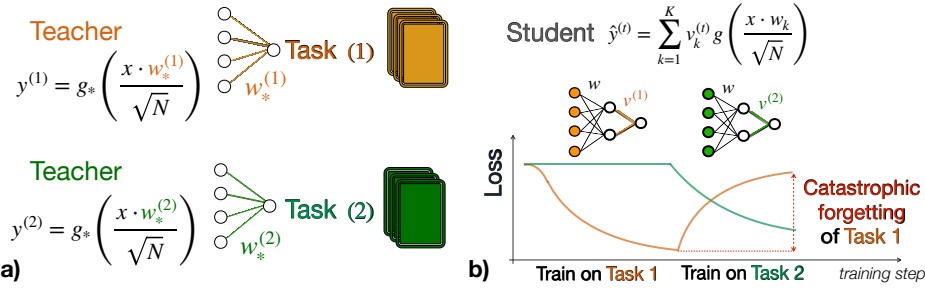

Figure 1: **Pictorial representation of the continual learning task in the teacher-student setting.** A "student" network is trained on i.i.d. inputs from two teacher networks, defining two different tasks (panel **a**). The student has sufficient capacity to learn both tasks. However, sequential training results in catastrophic forgetting, where the performance on a previously learned task significantly deteriorates when a new task is introduced (panel **b**). Parameters: $K = T = 2$.

Several other works have combined ideas from machine learning and optimal control. Notably, Han et al. (2019) interpreted deep learning as an optimal control problem on a dynamical system, where the control variables correspond to the network parameters. Chen & Hazan (2024) formulated meta-optimization as an optimal control problem, but their analysis did not involve dimensionality reduction techniques nor did it address task selection.

## 2 MODEL-BASED THEORETICAL FRAMEWORK

We adopt a model-based approach to investigate the supervised learning of multiple tasks. Following Lee et al. (2021; 2022), we consider a teacher-student framework (Gardner & Derrida, 1989). A "student" neural network is trained on synthetic inputs $\boldsymbol{x} \in \mathbb{R}^N$, drawn i.i.d. from a standard Gaussian distribution, $x_i \sim \mathcal{N}(0, 1)$. The labels for each task $t = 1, \dots, T$ are generated by single-layer "teacher" networks: $y^{(t)} = g_*(\boldsymbol{x} \cdot \boldsymbol{w}_*^{(t)}/\sqrt{N})$, where $\boldsymbol{W}_* = (\boldsymbol{w}_*^{(1)}, \dots, \boldsymbol{w}_*^{(T)})^\top \in \mathbb{R}^{T \times N}$ denote the corresponding teacher vectors, and $g_*$ the activation function. The student is a two-layer neural network with $K$ hidden units, first-layer weights $\boldsymbol{W} = (\boldsymbol{w}_1, \dots, \boldsymbol{w}_K)^\top \in \mathbb{R}^{K \times N}$, activation function $g$, and second-layer weights $\boldsymbol{v} \in \mathbb{R}^K$. It outputs the prediction:

$$\hat{y}(\boldsymbol{x}; \boldsymbol{W}, \boldsymbol{v}) = \sum_{k=1}^{K} v_k \, g\left(\frac{\boldsymbol{x} \cdot \boldsymbol{w}_k}{\sqrt{N}}\right) . \tag{1}$$

Following a standard *multi-headed* approach to continual learning (Zenke et al., 2017; Farquhar & Gal, 2018), we allow for task-dependent readout weights: $\boldsymbol{V} = (\boldsymbol{v}^{(1)}, \dots, \boldsymbol{v}^{(T)})^\top \in \mathbb{R}^{T \times K}$. Specifically, the readout for task $t$ is updated only when that task is presented. While the readout is switched during training according to the task under consideration, the first-layer weights are shared across tasks. A pictorial representation of this model is displayed in Fig. 1. Training is performed via Stochastic Gradient Descent (SGD) on the squared loss of $y^{(t)}$ and $\hat{y}^{(t)} = \hat{y}(\boldsymbol{x}; \boldsymbol{W}, \boldsymbol{v}^{(t)})$. We consider the *online* regime, where at each training step the algorithmic update is computed using a new sample $(\boldsymbol{x}, y^{(t)})$. The generalisation error of the student on task $t$ is given by

$$\varepsilon_t(\boldsymbol{W}, \boldsymbol{V}, \boldsymbol{W}_*) := \frac{1}{2}\left\langle \left(y^{(t)} - \hat{y}^{(t)}\right)^2 \right\rangle = \frac{1}{2}\mathbb{E}_{\boldsymbol{x}}\left[\left(g^*\left(\frac{\boldsymbol{w}_*^{(t)} \cdot \boldsymbol{x}}{\sqrt{N}}\right) - \hat{y}(\boldsymbol{x}; \boldsymbol{W}, \boldsymbol{v}^{(t)})\right)^2\right] . \tag{2}$$

The angular brackets $\langle \cdot \rangle$ denote the expectation over the input distribution for a given set of teacher and student weights. Crucially, the error depends on the input data only through the preactivations

$$\lambda_k := \frac{\boldsymbol{x} \cdot \boldsymbol{w}_k}{\sqrt{N}} , \quad k = 1, \dots, K, \qquad \text{and} \qquad \lambda_*^{(t)} := \frac{\boldsymbol{x} \cdot \boldsymbol{w}_*^{(t)}}{\sqrt{N}} , \quad t = 1, \dots, T. \tag{3}$$

Eq. 3 defines jointly Gaussian variables with zero mean and second moments given by

$$
\begin{aligned}
M_{kt} &:= \mathbb{E}_{\boldsymbol{x}} \left[ \lambda_k \lambda_*^{(t)} \right] = \frac{\boldsymbol{w}_k \cdot \boldsymbol{w}_*^{(t)}}{N} \ , \\
Q_{kh} &:= \mathbb{E}_{\boldsymbol{x}} \left[ \lambda_k \lambda_h \right] = \frac{\boldsymbol{w}_k \cdot \boldsymbol{w}_h}{N} \ , \\
S_{tt'} &:= \mathbb{E}_{\boldsymbol{x}} \left[ \lambda_*^{(t)} \lambda_*^{(t')} \right] = \frac{\boldsymbol{w}_*^{(t')} \cdot \boldsymbol{w}_*^{(t)}}{N} \ ,
\end{aligned}
\tag{4}
$$

called *overlaps* in the statistical physics literature. Therefore, the dynamics of the generalisation error is entirely captured by the evolution of the student readouts $\boldsymbol{V}$ and the overlaps. As shown in Lee et al. (2021; 2022), we can track the evolution of the generalisation error in the high-dimensional. We leverage this description and optimal control theory to derive *optimal training protocols* for multi-task learning. In particular, we optimise over task selection and learning rate.

***Forward* training dynamics.**  First, we derive equations governing the dynamics of the overlaps and readouts under a given task-selection protocol. These equations fully determine the evolution of the generalization error. For the remainder of the paper, we consider $K = T$ to guarantee that the student network has enough capacity to learn all tasks perfectly. Teacher vectors are normalised, and the task similarity is tuned by a parameter $\gamma$, so that $S_{tt'} = \delta_{t,t'} + \gamma(1 - \delta_{t,t'})$. For simplicity, it is useful to encode all the relevant degrees of freedom—namely, the overlaps and the readout weights—in the same vector. We use the shorthand notation $\mathbb{Q} = \left( \text{vec}(\boldsymbol{Q}), \text{vec}(\boldsymbol{M}), \text{vec}(\boldsymbol{V}) \right)^\top \in \mathbb{R}^{K^2 + 2KT}$. As further discussed in Appendix A, in the limit of large input dimension $N \to \infty$ with $K, T \sim \mathcal{O}_N(1)$, the training dynamics is described by a set of ODEs

$$
\frac{d\mathbb{Q}(\alpha)}{d\alpha} = f_{\mathbb{Q}} \left( \mathbb{Q}(\alpha), \boldsymbol{u}(\alpha) \right) \qquad \text{with } \alpha \in (0, \alpha_F] \ .
\tag{5}
$$

The parameter $\alpha$ denotes the effective training *time*—the ratio between training steps and input dimension $N$. The vector $\boldsymbol{u}$ encodes the dynamical variables that we want to control optimally. In particular, we study the optimal schedules for task-selection $t_c(\alpha)$ and learning rate $\eta(\alpha)$. Here, $t_c(\alpha) \in \{1, \ldots, T\}$ indicates on which task the student is trained at time $\alpha$. The specific form of the functions $f_{\mathbb{Q}}$ is derived in Appendix A. The initial condition $\mathbb{Q}(0)$ matches the initialisation of the SGD algorithm. In particular, the initial first-layer weights and readout weights are drawn i.i.d. from a normal distribution with variances of $10^{-3}$ and $10^{-2}$, respectively. Notably, the trajectory appears to be largely independent of the specific initialisation of the first-layer weights. For instance, in Fig. 2, simulations and theory correspond to different initialisations, yet the curves show excellent agreement. Let us stress that Eq. 5 is a low-dimensional deterministic equation that fully captures the high-dimensional stochastic dynamics of SGD as $N \to \infty$. This dimensionality reduction is crucial to apply the optimal control techniques presented in the next section.

**Optimal control framework and *backward* conjugate dynamics.**  Our first main contribution is to derive training strategies that are optimal with respect to the generalisation performance *at the end of training* and on *all tasks*. In practice, the goal of the optimisation process is to minimise a linear combination of the generalisation errors on the different tasks at the final training time $\alpha_F$:

$$
h(\mathbb{Q}(\alpha_F)) = \sum_{t=1}^{T} c_t \, \varepsilon_t(\mathbb{Q}(\alpha_F)) \qquad \text{with} \quad c_t \geq 0 \text{ and } \sum_{t=1}^{T} c_t = 1 \ ,
\tag{6}
$$

where the coefficients $c_t$ identify the relative importance of different tasks and $\varepsilon_t$ denotes the infinite-dimensional limit of the average generalisation error on task $t$, as defined in Eq. 2. Crucially, we have an analytic expression for $\varepsilon_t$, derived in Appendix A. In the remainder of the paper, we assume equally important tasks $c_t = 1/T$. As customary in optimal control theory (Pontryagin, 1957), we adopt a variational approach to solve the problem. We define the cost functional

$$
\mathcal{F}[\mathbb{Q}, \hat{\mathbb{Q}}, \boldsymbol{u}] = h\left(\mathbb{Q}(\alpha_F)\right) + \int_0^{\alpha_F} d\alpha \, \hat{\mathbb{Q}}(\alpha)^\top \left[ -\frac{d\mathbb{Q}(\alpha)}{d\alpha} + f_{\mathbb{Q}}\left(\mathbb{Q}(\alpha), \boldsymbol{u}(\alpha)\right) \right] \ ,
\tag{7}
$$

where the *conjugate order parameters* $\hat{\mathbb{Q}} = \left( \text{vec}(\hat{\boldsymbol{Q}}), \text{vec}(\hat{\boldsymbol{M}}), \text{vec}(\hat{\boldsymbol{V}}) \right)^\top$ enforce the training dynamics in the training interval $\alpha \in [0, \alpha_F]$. Finding the optimal protocol amounts to minimising the

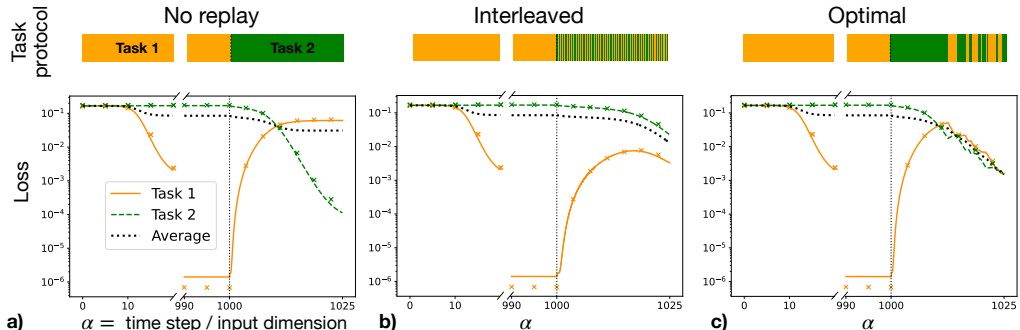

Figure 2: **How to learn a new task without forgetting the old one?** The student is trained on task 1 until convergence during the first phase ($\alpha \in [0, 1000]$), then task 2 is introduced. During the second phase ($\alpha \in (1000, 1025]$), task 1 may be replayed to prevent forgetting. For better visibility, we only display the regions $\alpha \in [0, 20] \cup [990, 1025]$. We compare three strategies: **a)** no replay, **b)** interleaved replay, i.e., alternating between the two tasks, **c)** the optimal strategy derived in Sec. 2. Crosses mark numerical simulations of a single trajectory at $N = 20000$, lines mark the solution of Eq. 5. Colour bars represent the protocol $t_c$. Parameters: $\gamma = 0.3$, $K = T = 2$, and $\eta = 1$.

cost functional $\mathcal{F}$ with respect to $\mathbb{Q}$, $\hat{\mathbb{Q}}$, and $\boldsymbol{u}$. We defer the details of this variational procedure to Appendix A, and present only the main steps here. For a general introduction to the control methods adopted here, see, e.g., Kirk (2004). The minimisation with respect to $\mathbb{Q}$ provides a set of equations for the *backward* dynamics of the conjugate parameters

$$-\frac{\mathrm{d}\hat{\mathbb{Q}}(\alpha)^{\top}}{\mathrm{d}\alpha} = \hat{\mathbb{Q}}(\alpha)^{\top}\nabla_{\mathbb{Q}}f_{\mathbb{Q}}(\mathbb{Q}(\alpha), \boldsymbol{u}(\alpha)) \qquad \text{with } \alpha \in [0, \alpha_F). \tag{8}$$

The final condition for the dynamics is given by

$$\hat{\mathbb{Q}}(\alpha_F) = \nabla_{\mathbb{Q}}h(\mathbb{Q}_F) = \sum_{t=1}^{T} c_t \nabla_{\mathbb{Q}}\varepsilon_t(\mathbb{Q}(\alpha_F)). \tag{9}$$

The optimal control curve $\boldsymbol{u}^*(\alpha)$ is obtained as the solution of the minimisation:

$$\boldsymbol{u}^*(\alpha) = \operatorname*{argmin}_{\boldsymbol{u} \in \mathcal{U}} \left\{ \hat{\mathbb{Q}}(\alpha)^{\top} f_{\mathbb{Q}}(\mathbb{Q}(\alpha), \boldsymbol{u}) \right\}, \tag{10}$$

where $\mathcal{U}$ is the set of allowed controls, that can be either continuous or discrete. For instance, for task selection we take $u(\alpha) = t_c(\alpha)$ and $\mathcal{U} = \{1, 2 \ldots, T\}$, where we use the notation $t_c(\alpha)$ to indicate the current task, i.e., the task on which the student is trained at time $\alpha$. When optimising over both task selection and learning rate schedule we take $\boldsymbol{u} = (t_c, \eta)$ and $\mathcal{U} = \{1, 2 \ldots, T\} \times \mathbb{R}^+$. Crucially, the optimal control equations 5, 8, and 10 must be iterated until convergence, starting from an initial guess on $\boldsymbol{u}$, which can be for instance taken at random. Let us stress that the space $\mathcal{U}$ of possible controls is high-dimensional and hence it is not feasible to explore it via greedy search strategies.

## 3 RESULTS AND APPLICATIONS

The theoretical framework of Sec. 2 is extremely flexible and can be applied to a variety of settings. We will present a technical paper focused on the method in a specialised venue. In this section, we focus on specific settings and investigate in detail the impact of optimal training protocols on both synthetic and real tasks. First, we consider the synthetic teacher-student framework. We compute optimal task-selection protocols, investigating their structure and interplay with task similarity and learning rate schedule. Then, we transfer the insights gained from the interpretation of the optimal strategies in synthetic settings to applications on real datasets.

### 3.1 EXPERIMENTS ON SYNTHETIC DATA

We formulate the problem of continual learning as follows. During a first training phase, the student learns perfectly task $t = 1$. Then, the goal is to learn a new task $t = 2$ without forgetting the

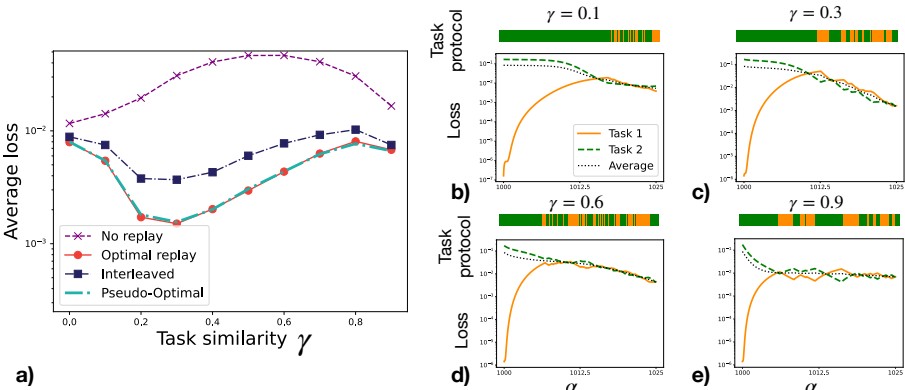

Figure 3: **The impact of task similarity on continual learning. a)** Average loss on both tasks at the end of the second training phase as a function of the task similarity $\gamma$ under the replay setting from Fig. 2. Different lines correspond to different strategies: no replay (purple crosses), optimal replay (red dots), interleaved (blue squares), pseudo-optimal replay (cyan dashed line). **b-e)** Optimal replay strategies for different values of $\gamma = 0.1, 0.3, 0.6, 0.9$. Colour bars represent the protocol $t_c(\alpha)$.

previous one. A given time window of duration $\alpha_F$ is assigned for the second training phase. In particular, we investigate the role of *replay*—i.e., presenting samples from task 1 during the second training phase—and the structure of the optimal replay strategy.

We use the equations derived in Sec. 2 to study optimal replay during the second phase of training. To this end, we take the task-selection variable as our control $u(\alpha) = t_c(\alpha) \in \{1, 2\}$, while we set $t_c = 1$ during the first training phase. The result of the optimisation in Eq. 10 strikes the balance between training on the new task and replaying the old task. We do not enforce any constraints on the number of samples from task 1 to use in the second phase. Therefore, our method provides both the optimal *fraction* of replayed samples and the optimal task *ordering*, depending on the time window $\alpha_F$. Fig. 2 compares the learning dynamics of three different strategies, depicting the loss on task 1 (full orange line), task 2 (dashed green line), and their average (dotted black line) as a function of the training time $\alpha$. Numerical simulations—marked by cross symbols—are in excellent agreement with our theory. Deviations are smaller than $1/\sqrt{N}$, compatible with finite-size effects.

The student is trained exclusively on task 1 until $\alpha = 1000$, when the task is perfectly learned with loss $\sim 10^{-6}$. Then, the student is trained on a combination of new and old tasks for a training time of duration $\alpha_F = 25$. A colour bar above each plot illustrates the associated task-selection strategy $t_c(\alpha)$. Panel **a)** shows training without replay, where only task 2 is presented in the second phase. We observe catastrophic forgetting of task 1. Panel **b)** shows a heuristic "interleaved" strategy, where training alternates one sample from the new task to one sample from the old one. As observed in Lee et al. (2022), the interleaved strategy already provides a performance gain, demonstrating the relevance of replay to mitigate catastrophic forgetting. Panel **c)** of Fig. 2 shows the loss dynamics for the optimal replay strategy. Notably, the optimal strategy exhibits a complex structure and displays a clear performance improvement over the other two strategies. In particular, we find that the optimal task-selection strategy always presents an initial phase where training is performed only on the new task. This behaviour is observed consistently across a range of task similarities.

**The impact of task similarity.** To understand the structure of the optimal strategy, we examine its performance in relation to task similarity $\gamma$. Fig. 3**a)** depicts the average loss at the end of training as a function of $\gamma$. For the no-replay strategy, we reproduce the findings from previous works (Lee et al., 2021; 2022): the highest error occurs at intermediate task similarity. Lee et al. (2022) explained this non-monotonicity as a trade-off between node re-use and node activation. Specifically, for small $\gamma$, there is minimal interference between tasks. One hidden neuron predominantly aligns with the new task, while the other neuron retains the knowledge of the old task, leading to task *specialisation*. At large $\gamma$, features from task 1 are reused for task 2, avoiding forgetting. However, at intermediate $\gamma$, interference is maximal: both neurons quickly align with task 2, and task 1 is forgotten. Remarkably,

Fig. 3**a)** shows that replay reverses this trend, with the minimal error occurring at intermediate $\gamma$. To explain this nontrivial behaviour, we must first understand the optimal replay protocol.

**Interpretation of the optimal replay structure.** The optimal replay dynamics is illustrated in panels **b-e)** of Fig. 3 and displays a highly structured protocol. We can interpret this structure a posteriori: an initial *focus phase* without replay is followed by a *revision phase* involving interleaved replay. The transition between these two phases corresponds approximately to the point at which the loss on the new task matches the loss on the old one. To investigate the significance of this structure, we also test an interleaved strategy, plotted in Fig. 3**a)**. In this case, the task ordering in the second training phase is fully randomised while maintaining the same overall replay fraction of the optimal strategy. This protocol has a performance gap compared to the optimal one, showing the importance of a properly structured replay scheme. Additionally, we test a "pseudo-optimal" variant, where the *focus phase* is retained, but the *revision phase* is randomised. This variant performs comparably to the optimal strategy, suggesting that while the specific order of the revision phase is largely unimportant, it is key to precede it with a training phase on the new task.

We can now attempt to understand the inverted non-monotonic behaviour of the average loss as a function of $\gamma$ under the optimal protocol. First, as shown in Fig. 8 of Appendix C, the optimal protocol achieves a good level of node specialisation across all values of $\gamma$. Thus, replay prevents the task interference that typically causes performance deterioration at intermediate $\gamma$. The non-monotonic behaviour of the optimal curve in Fig. 3**a)** arises from a different origin, involving two opposing effects related to the first-layer weights and the readout. The initial decrease of the loss with $\gamma$ is quite intuitive, as only minimal knowledge can be transferred from task 1 to task 2 when $\gamma$ is small. Consequently, the focus phase on task 2 must be longer for smaller $\gamma$, leaving less time to revise task 1, thereby reducing performance. On the other hand, the performance decrease observed in Fig. 3a) for $\gamma > 0.3$ is more subtle and is related to the readout layer. Once the two hidden neurons have specialised—each aligning with one of the teacher vectors—we expect the readout weights corresponding to the incorrect teacher to be suppressed. Specifically, if $\boldsymbol{w}_1 = \boldsymbol{w}_*^{(1)}$ and $\boldsymbol{w}_2 = \boldsymbol{w}_*^{(2)}$, the learning dynamics should drive the readout weights $\boldsymbol{v}^{(1)} = (v_1^{(1)}, v_2^{(1)})^\top$ and $\boldsymbol{v}^{(2)} = (v_1^{(2)}, v_2^{(2)})^\top$ towards $\boldsymbol{v}^{(1)} = (1,0)^\top$ and $\boldsymbol{v}^{(2)} = (0,1)^\top$ to achieve full recovery of the teacher networks. As shown in Fig. 8 of Appendix C, the time required to suppress the off-diagonal weights $v_2^{(1)}$ and $v_1^{(2)}$ increases as $\gamma \to 1$. This is intuitive, as higher task similarity $\gamma$ reduces the distinction between tasks, slowing the suppression of the off-diagonal weights. In Appendix B, we derive analytically the convergence timescale $\alpha_{\mathrm{conv}}$ of the readout layer, showing that

$$\alpha_{\mathrm{conv}} = \frac{3\pi}{\eta(\pi - 6\arcsin(\gamma/2))}, \qquad (11)$$

where $\eta$ is the learning rate. This timescale is a monotonically increasing function of $\gamma$ and diverges as $\gamma \to 1$ with $\alpha_{\mathrm{conv}} \approx \sqrt{3}\pi/(2\eta(1-\gamma))$. This result explains the performance decrease of the optimal strategy as $\gamma \to 1$. In summary, the performance decrease for $\gamma \to 0$ is due to the first-layer weights, while for $\gamma \to 1$ it is related to the readout weights.

**Optimal learning rate schedules.** While the above results are obtained with a constant learning rate, annealing schedules are widely used in machine learning. Thus, it is relevant to study the optimal interplay between replay and learning rate. Optimal learning rate dynamics have been studied with a similar approach in Saad & Rattray (1997). We jointly optimise the task-selection protocol together with the learning rate to investigate its impact on continual learning. Fig. 4 shows the optimal learning rate schedule for task similarity $\gamma = 0.3$ in the second training

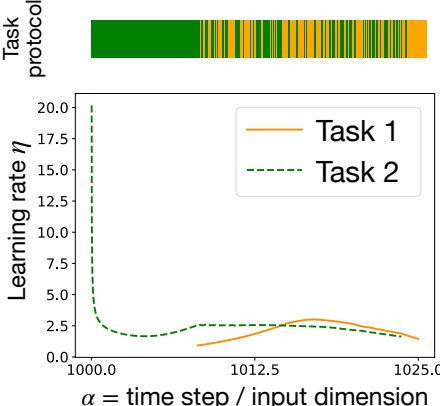

Figure 4: **Joint optimisation of task-selection and learning-rate.** Optimal learning rate as a function of training time $\alpha$ for the same parameters as Fig. 2. There is a single optimal learning rate curve, but for visibility purposes we show it as a solid orange line when training on task 1 and a dashed green line on task 2. The task-selection protocol $t_c(\alpha)$ is illustrated in the colour bar.

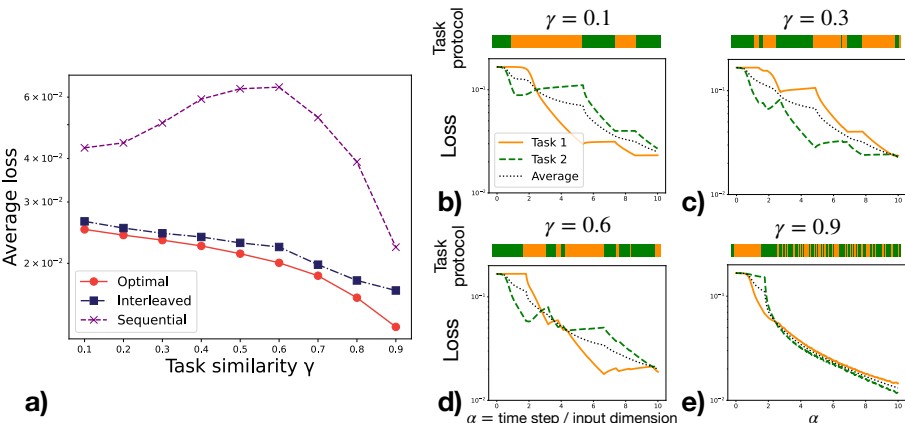

Figure 5: **The impact of task similarity on multi-task learning. a)** Average loss as a function of task similarity $\gamma$ at the end of training ($\alpha_F = 10$). Different lines correspond to different strategies: sequential (purple crosses), optimal (red dots), and randomly interleaved with $50\%$ samples from each task (blue squares). **b-e)** Optimal replay strategy for different values of $\gamma = 0.1, 0.3, 0.6, 0.9$. Parameters: $K\!=\!T\!=\!2$ and $\eta\!=\!10$.

phase of duration $\alpha_F = 25$. Similarly as for constant learning rate (panel **c)** of Fig. 3), optimal task-selection is characterised by an initial focus phase. Notably, this phase coincides with a strong annealing of the learning rate to achieve optimal performance. Intuitively, when learning the new task, the learning rate starts high and is gradually decreased over time. Interestingly, while entering the revision phase, the optimal learning rate schedule exhibits a highly nontrivial structure (see Fig. 4). Indeed, although the optimal learning rate curve is unique, we find that effectively it can be seen as two different curves, associated to the respective tasks. In practice, the optimal learning rate curve "jumps" between these two curves according to the task selected at a given training time. performance. Fig. 9 of Appendix C shows that the joint optimization of learning rate and task selection outperforms all other protocols, including exponential and power-law learning rate schedules combined with interleaved replay.

**Multi-task learning from scratch.**  We also consider a multi-task setting (see Fig. 5) where both tasks must be learned from scratch within a fixed number of steps, corresponding to a total training time $\alpha_F$. We first consider sequential learning, i.e., training only on task 1 for $\alpha < \alpha_F/2$, then only on task 2, or vice versa. As shown in Fig. 5**a)** sequential learning leads to catastrophic forgetting, with the worst performance observed at intermediate task similarity. In contrast, a randomly interleaved strategy, where examples from both tasks are presented in equal proportion but in random order, shows significant improvement. This approach can exploit task similarity, leading to a monotonic decrease in average loss as $\gamma$ increases. The optimal strategy, displayed in Fig. 5**b-e)** for various values of $\gamma$, follows a structured interleaved protocol that further enhances performance. Contrary to the continual learning framework, the optimal structure gives only marginal gain over the plain interleaved strategy. This observation aligns with our pseudo-optimal strategy, which suggests employing interleaved replay once performance on both tasks becomes comparable.

## 3.2 EXPERIMENTS ON REAL DATA

We consider the experimental framework established in Ramasesh et al. (2020); Lee et al. (2022) for the study of task similarity in relation to catastrophic forgetting. We use the Fashion-MNIST dataset (Xiao et al., 2017) to generate upstream and downstream tasks. The upstream dataset— $\mathcal{D}_1 = \{\boldsymbol{x}_i^{(1)}, y_i^{(1)}\}_i$—consists in a pair of classes from the standard dataset. The downstream dataset is generated by a linear interpolation of the upstream dataset with a second auxiliary dataset— $\tilde{\mathcal{D}} = \{\tilde{\boldsymbol{x}}_i, \tilde{y}_i\}_i$—containing a new pair of classes,

$$\mathcal{D}_2 = \{\boldsymbol{x}_i^{(2)}, y_i^{(2)}\}_i = \{\gamma \boldsymbol{x}_i^{(1)} + (1-\gamma)\tilde{\boldsymbol{x}}_i, \gamma y_i^{(1)} + (1-\gamma)\tilde{y}_i\}_i, \quad (12)$$

where the parameter $\gamma$ controls the task similarity. We then train a standard two-layer feedforward ReLU neural network on the two datasets using online SGD on a squared error loss. We consider a dynamical multi-head architecture (Zhou et al., 2012; Rusu et al., 2016) where the readout weights are changed switching from one task to another, but the hidden layer is shared. During training, we apply the three strategies discussed in the previous sections: a no-replay strategy, a strategy with interleaved replay, and a "pseudo-optimal" strategy. Recall that the latter is inspired by the optimal protocol derived in the previous section. It consists of an initial phase of training exclusively on the new task until performance on both tasks becomes comparable, followed by a phase of interleaved replay. Crucially, this protocol can be easily implemented in practice, as it only requires an estimate of the generalisation error on the two tasks, which can be obtained in real-world settings.

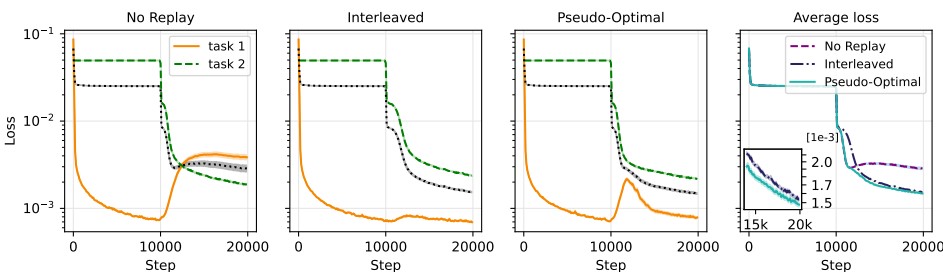

Figure 6: **Training dynamics.** Training curves on the modified fashion MNIST task at similarity $\gamma = 0.5$. The network is trained for $10.000$ steps on the first task before switching to the second task and being trained for additional $10.000$ steps. The results are obtained from $100$ realisations of the problem. The first three panels show the test loss on task 1 (solid orange), task 2 (dashed green), and their average (dotted black) for three training strategies, from left to right: no-replay, interleaved, and pseudo-optimal. The rightmost panel shows the average loss over the entire training.

Fig. 6 shows the training loss under the different training protocols for $\gamma = 0.5$. While the no-replay strategy appears to be successful for small downstream datasets (i.e., a few training steps in the online framework) in the longer run it leads to strong forgetting and high average loss. This behavior is intuitive: for small datasets, the initial loss on the new task is high, leading to a substantial decrease in loss early on, which temporarily outweighs the decline in performance on the previous task. The interleaved is beneficial in the long run but largely slows down learning of the new task. Overall, the pseudo-optimal protocol identified in Sec. 3.1 shows a better performance over the entire trajectory.

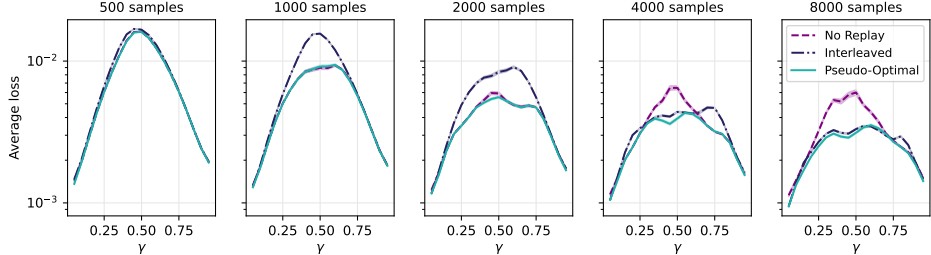

Figure 7: **Average loss comparison.** The figure focuses on the average loss and shows the final loss achieved by the three strategies as we increase the size of task 2 (from left to right: $500$, $1.000$, $2.000$, $4.000$, and $8.000$ samples) while task 1 has always $10.000$ samples. Individual panels show the performance of the three strategies as we span the value of $\gamma$ form $0.05$ to $0.95$.

This result is not limited to the specific value of $\gamma$. Fig. 7 shows the average final loss as a function of $\gamma$. From left to right, different panels correspond to an increasing number of available samples for the downstream task, comparable to $\alpha_F$ in the theoretical model. For small downstream tasks, the no-replay strategy is optimal, as shown in the second and third panels. As the size of the downstream task increases, the interleaved strategy approaches optimal performance while the no-replay becomes suboptimal. The pseudo-optimal strategy combines the advantages of both approaches, in-

terpolating between no-replay and interleaved strategies to automatically adapt to the computational budget. This results in the best overall performance across regimes. We confirm this observation with additional experiments on CIFAR10 in Appendix C.2. Notably, despite the differences between the synthetic and real settings—such as data structure—the pseudo-optimal strategy remains effective on real-world data, demonstrating its robustness and broad applicability.

## 4  DISCUSSION

**Conclusion.**  In this work, we introduce a systematic approach for identifying and interpreting optimal task-selection strategies in synthetic learning settings. We consider a teacher-student scenario as a prototypical continual learning problem to achieve analytic understanding of supervised multi-task learning. We incorporate prior results on exact ODEs for high-dimensional online SGD dynamics into a control-theory framework that allows us to derive exact equations for the optimal protocols. Our theory reveals that optimal task-selection protocols are typically highly structured—alternating between focused learning and interleaved replay phases—and display a nontrivial interplay with task similarity. We also identify highly structured optimal learning rate schedules that synchronise with optimal task-selection to enhance overall performance. Finally, leveraging insights from the synthetic setting, we extract a pseudo-optimal strategy applicable to real tasks.

**Limitations and Perspectives.**  This work takes a first step toward understanding the theory behind optimal training protocols for neural networks. In the following, we discuss current limitations and outline promising directions for future research. First, Pontryagin's maximum principle provides a necessary condition for optimality but does not guarantee a global optimum. Nevertheless, the strategies derived from this approach in the settings under consideration perform significantly better than previously proposed heuristics. Additionally, Pontryagin's principle does not easily extend to stochastic problems. This limitation is overcome in the high-dimensional limit where concentration results provide deterministic dynamical equations. For simplicity, we focus on i.i.d. Gaussian inputs, but our analysis can be extended to more structured data models (Goldt et al., 2020; Loureiro et al., 2021; Adomaityte et al., 2023) to study how input distribution affects task selection. In particular, we do not model the relative task difficulty—an important extension that naturally connects to the theory of curriculum learning (Weinshall & Amir, 2020; Saglietti et al., 2022; Cornacchia & Mossel, 2023; Abbe et al., 2023). Furthermore, it would be interesting to go beyond the study of online dynamics to understand the impact of memorisation in batch learning settings (Sagawa et al., 2020). Existing results in the spurious correlations (Ye et al., 2021) and fairness (Ganesh et al., 2023) literature suggest a strong dependence of the classifier's bias on the presentation order in batch learning. Our method can be applied to mean-field models—like (Mannelli et al., 2024; Jain et al., 2024)—to theoretically investigate this phenomenon. An interesting extension of our work involves applying recently-developed statistical physics methods to the study of deeper networks and more complex learning architectures (Bordelon & Pehlevan, 2022b;a; Rende et al., 2024; Tiberi et al., 2024). Another interesting direction concerns finding optimal protocols for shaping, where task order significantly impacts both animal learning and neural networks (Skinner, 1938; Tong et al., 2023; Lee et al., 2024).

### REPRODUCIBILITY STATEMENT

The code to reproduce the numerical experiments is available in the supplementary material.

### ACKNOWLEDGMENT

We thank Rodrigo Carrasco-Davis, Sebastian Goldt, and Andrew Saxe for useful feedback. This work was supported by a Leverhulme Trust International Professorship grant [number LIP-2020-014], the Wallenberg AI, Autonomous Systems, and Software Program (WASP), and by the Simons Foundation (Award Number: 1141576).

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

## A   DETAILS ON THE THEORETICAL DERIVATIONS

In this appendix, we provide detailed derivations of the equations in Sec. 2 of the main text. In the interest of completeness, we also report the derivation of the ODEs describing online SGD dynamics and the generalisation error as a function of the order parameters, first derived in (Lee et al., 2021). We remind that inputs are $N-$dimensional vectors $\boldsymbol{x} \in \mathbb{R}^N$ with independent identically distributed (i.i.d.) standard Gaussian entries $x_i \sim \mathcal{N}(0, 1)$, while the labels are generated by single-layer teacher networks: $y^{(t)} = g_*(\boldsymbol{x} \cdot \boldsymbol{w}_*^{(t)}/\sqrt{N})$, $t = 1, \ldots, T$, with a different teacher for each task. The student is a one-hidden layer network that outputs the prediction:

$$\hat{y}^{(t)} = \sum_{k=1}^{K} v_k^{(t)} g\left(\frac{\boldsymbol{x} \cdot \boldsymbol{w}_k}{\sqrt{N}}\right) . \tag{13}$$

In the main text, we focus on the case $K = T$, where the student has, in principle, the capacity to perfectly solve the problem and represent all teachers. Specifically, there exists a configuration of the student's parameters that achieves perfect recovery. This configuration corresponds to aligning each of the student's hidden neurons with a specific teacher/task. Explicitly, this configuration is given by $\boldsymbol{w}_k = \boldsymbol{w}_*^{(k)}$ and $v_k^{(t)} = \delta_{k,t}$, where $\delta_{k,t}$ denotes the Kronecker delta. However, our theory remains valid for arbitrary $K$ and $T$.

We focus on the *online* (on *one-pass*) setting, so that at each training step the student network is presented with a fresh example $\boldsymbol{x}^\mu$, $\mu = 1, \ldots, P$, and $P/N \sim \mathcal{O}_N(1)$. The weights of the student are updated through gradient descent on $\frac{1}{2}(\hat{y}^{(t)} - y^{(t)})^2$ following the task-selection protocol $t_c$:

$$\boldsymbol{w}_k^{\mu+1} = \boldsymbol{w}_k^\mu - \eta^\mu \Delta^{(t_c)\mu} v_k^{(t_c)\mu} g'\left(\lambda_k^\mu\right) \frac{\boldsymbol{x}^\mu}{\sqrt{N}} ,$$

$$v_k^{(t)\mu+1} = v_k^{(t)\mu} - \frac{\eta^\mu}{N} \Delta^{(t)\mu} g(\lambda_k^\mu)\delta_{t,t_c} , \tag{14}$$

$$\Delta^{(t)\mu} := \hat{y}^{(t)\mu} - y^{(t)\mu} = \sum_{k=1}^{K} v_k^{(t)} g(\lambda_k^\mu) - g_*(\lambda_*^{(t)\mu}) ,$$

where $\eta^\mu$ denotes the (possibly time-dependent) learning rate and we have rescaled it by $N$ in the dynamics of the readout weights for future convenience. We have defined the preactivations, a.k.a. *local fields*,

$$\lambda_k^\mu := \frac{\boldsymbol{x}^\mu \cdot \boldsymbol{w}_k^\mu}{\sqrt{N}} , \qquad\qquad \lambda_*^{(t)\mu} := \frac{\boldsymbol{x}^\mu \cdot \boldsymbol{w}_*^{(t)}}{\sqrt{N}} . \tag{15}$$

Notice that, due to the online-learning setup, at each training step the input $\boldsymbol{x}$ is independent of the weights. Therefore, due to the Gaussianity of the inputs, the local fields are also jointly Gaussian with zero mean and second moments given by the *overlaps*:

$$
\begin{aligned}
M_{kt} &:= \mathbb{E}_{\boldsymbol{x}}\left[\lambda_k \lambda_*^{(t)}\right] = \frac{\boldsymbol{w}_k \cdot \boldsymbol{w}_*^{(t)}}{N} \ , \\
Q_{kh} &:= \mathbb{E}_{\boldsymbol{x}}\left[\lambda_k \lambda_h\right] = \frac{\boldsymbol{w}_k \cdot \boldsymbol{w}_h}{N} \ , \\
S_{tt'} &:= \mathbb{E}_{\boldsymbol{x}}\left[\lambda_*^{(t)} \lambda_*^{(t')}\right] = \frac{\boldsymbol{w}_*^{(t')} \cdot \boldsymbol{w}_*^{(t)}}{N} \ .
\end{aligned}
\tag{16}
$$

## A.1 GENERALISATION ERROR AS A FUNCTION OF THE ORDER PARAMETERS

We can write the generalisation error (Eq. 2 of the main text) as an average over the local fields:

$$
\begin{aligned}
\varepsilon_t\left(\boldsymbol{W}, \boldsymbol{V}, \boldsymbol{W}_*\right) = &\frac{1}{2}\sum_{k,h} v_k^{(t)} v_h^{(t)} \mathbb{E}_{\boldsymbol{\lambda}, \boldsymbol{\lambda}_*}\left[g(\lambda_k)g(\lambda_h)\right] + \frac{1}{2}\mathbb{E}_{\boldsymbol{\lambda}, \boldsymbol{\lambda}_*}\left[g_*(\lambda_*^{(t)})^2\right] \\
&- \sum_k v_k^{(t)} \mathbb{E}_{\boldsymbol{\lambda}, \boldsymbol{\lambda}_*}\left[g(\lambda_k)g_*(\lambda_*^{(t)})\right] \ .
\end{aligned}
\tag{17}
$$

where the expectation is computed over the multivariate Gaussian distribution

$$
\begin{aligned}
P(\boldsymbol{\lambda}, \boldsymbol{\lambda}_*) &= \frac{1}{\sqrt{(2\pi)^{K+T}|\boldsymbol{C}|}}\exp\left(-\frac{1}{2}(\boldsymbol{\lambda}, \boldsymbol{\lambda}_*)^\top \boldsymbol{C}^{-1}(\boldsymbol{\lambda}, \boldsymbol{\lambda}_*)\right) \ , \\
\boldsymbol{C} &= \begin{pmatrix} \boldsymbol{Q} & \boldsymbol{M} \\ \boldsymbol{M}^\top & \boldsymbol{S} \end{pmatrix} \ .
\end{aligned}
\tag{18}
$$

From now on, we adopt the unified notation

$$
I_2(\beta, \rho) := \mathbb{E}_{\boldsymbol{\lambda}, \boldsymbol{\lambda}_*}\left[g_\beta(\lambda_\beta)g_\rho(\lambda_\rho)\right] \ ,
\tag{19}
$$

where $\beta, \rho$ can refer both to the indices of the student weights $k, h$ or the tasks $t, t'$. We can then rewrite the generalisation error as

$$
\varepsilon_t\left(\boldsymbol{W}, \boldsymbol{V}, \boldsymbol{W}_*\right) = \frac{1}{2}\sum_{k,h} v_k^{(t)} v_h^{(t)} I_2(k, h) + \frac{1}{2}I_2(t, t) - \sum_k v_k^{(t)} I_2(k, t) \ .
\tag{20}
$$

In all the results presented in Sec. 3, we consider $g(z) = g_*(z) = \mathrm{erf}\left(z/\sqrt{2}\right)$. In this case, there is an analytic expression for the integral $I_2$ (Saad & Solla, 1995a):

$$
I_2(\beta, \rho) = \frac{2}{\pi}\arcsin\frac{q_{\beta\rho}}{\sqrt{1+q_{\beta\beta}}\sqrt{1+q_{\rho\rho}}} \ ,
\tag{21}
$$

and we use the symbol $q$ to denote generically an overlap from Eq. 16, according to the choice of indices $\beta, \rho$, e.g., $q_{kh} = Q_{kh}$, $q_{kt} = M_{kt}$, and $q_{tt_c} = S_{tt_c}$. In this special case, the generalisation error can be written explicitly as a function of the overlaps

$$
\begin{aligned}
\varepsilon_t\left(\boldsymbol{W}, \boldsymbol{V}, \boldsymbol{W}_*\right) = &\frac{1}{\pi}\sum_{k,h} v_k^{(t)} v_h^{(t)} \arcsin\frac{Q_{kh}}{\sqrt{1+Q_{kk}}\sqrt{1+Q_{hh}}} + \frac{1}{\pi}\arcsin\frac{S_{tt}}{1+S_{tt}} \\
&- \frac{2}{\pi}\sum_k v_k^{(t)} \arcsin\frac{M_{kt}}{\sqrt{1+Q_{kk}}\sqrt{1+S_{tt}}} \ .
\end{aligned}
\tag{22}
$$

## A.2 ORDINARY DIFFERENTIAL EQUATIONS FOR THE FORWARD TRAINING DYNAMICS

Given that the generalisation error depends only on the overlaps, in order to characterise the learning curves we need to compute the equations of motion for the overlaps from the SGD dynamics of the weights given in Eq. 14. The order parameter $S_{tt'}$ associated to the teachers is constant in time. We obtain an ODE for $M_{kt}$ by multiplying both sides of the first of Eq. 14 by $\boldsymbol{w}_*^{(t)}$ and dividing by $N$:

$$
\frac{\boldsymbol{w}_k^{\mu+1} \cdot \boldsymbol{w}_*^{(t)}}{N} - \frac{\boldsymbol{w}_k^\mu \cdot \boldsymbol{w}_*^{(t)}}{N} = -\frac{\eta^\mu}{N}\Delta^{(t_c)\mu} v_k^{(t_c)\mu} g'(\lambda_k^\mu)\lambda_*^{(t)\mu} \ ,
\tag{23}
$$

where we stress the difference between $t_c$, the task selected for training at step $\mu$, and $t$, the task for which we compute the overlap. We define a "training time" $\alpha = \mu/N$ and take the infinite-dimensional limit $N \to \infty$. The parameter $\alpha$ becomes continuous and $M_{kt}$ concentrates to the solution of the following ODE:

$$\frac{\mathrm{d}M_{kt}}{\mathrm{d}\alpha} = -\eta v_k^{(t_c)} \mathbb{E}_{\boldsymbol{\lambda}, \boldsymbol{\lambda}_*} \left[ \Delta^{(t_c)} g'(\lambda_k) \lambda_*^{(t)} \right] := f_{\boldsymbol{M}, kt} \,, \tag{24}$$

where the expectation is computed over the distribution in Eq. 18. The ODE for $Q_{kh}$ is obtained similarly from Eq. 14:

$$\frac{\boldsymbol{w}_k^{\mu+1} \cdot \boldsymbol{w}_h^{\mu+1}}{N} - \frac{\boldsymbol{w}_k^{\mu} \cdot \boldsymbol{w}_h^{\mu}}{N} = -\frac{\eta^{\mu}}{N} \Delta^{(t_c)\mu} v_k^{(t_c)\mu} g'(\lambda_k^{\mu}) \lambda_h^{\mu} - \frac{\eta^{\mu}}{N} \Delta^{(t_c)\mu} v_h^{(t_c)\mu} g'(\lambda_h^{\mu}) \lambda_k^{\mu}$$
$$+ (\eta^{\mu})^2 \left( \Delta^{(t_c)\mu} \right)^2 v_k^{(t_c)\mu} v_h^{(t_c)\mu} g'(\lambda_k^{\mu}) g'(\lambda_h^{\mu}) \frac{\boldsymbol{x} \cdot \boldsymbol{x}}{N} \,. \tag{25}$$

In the infinite-dimensional limit, we find

$$\frac{\mathrm{d}Q_{kh}}{\mathrm{d}\alpha} = -\eta v_k^{(t_c)} \mathbb{E}_{\boldsymbol{\lambda}, \boldsymbol{\lambda}_*} \left[ \Delta^{(t_c)} g'(\lambda_k^{\mu}) \lambda_h^{\mu} \right] - \eta v_h^{(t_c)} \mathbb{E}_{\boldsymbol{\lambda}, \boldsymbol{\lambda}_*} \left[ \Delta^{(t_c)} g'(\lambda_h^{\mu}) \lambda_k^{\mu} \right] \tag{26}$$

$$+ \eta^2 v_k^{(t_c)} v_h^{(t_c)} \mathbb{E}_{\boldsymbol{\lambda}, \boldsymbol{\lambda}_*} \left[ \left( \Delta^{(t_c)} \right)^2 g'(\lambda_k) g'(\lambda_h) \right] := f_{\boldsymbol{Q}, kh} \,. \tag{27}$$

Finally, taking the infinite dimensional limit of the second Eq. 14, we find the ODE for the readout:

$$\frac{\mathrm{d}v_k^{(t)}}{\mathrm{d}\alpha} = -\eta \, \mathbb{E}_{\boldsymbol{\lambda}, \boldsymbol{\lambda}_*} \left[ \Delta^{(t)} g(\lambda_k) \right] \delta_{t, t_c} := f_{\boldsymbol{V}, tk} \,. \tag{28}$$

It is useful to write this system of ODEs in a more compact form. With the shorthand notation $\mathbb{Q} = (\mathrm{vec}(\boldsymbol{Q}), \mathrm{vec}(\boldsymbol{M}), \mathrm{vec}(\boldsymbol{V}))^{\top}$, $f_{\mathbb{Q}} = (\mathrm{vec}(f_{\boldsymbol{Q}}), \mathrm{vec}(f_{\boldsymbol{M}}), \mathrm{vec}(f_{\boldsymbol{V}}))^{\top}$, we can write

$$\frac{\mathrm{d}\mathbb{Q}(\alpha)}{\mathrm{d}\alpha} = f_{\mathbb{Q}} \left( \mathbb{Q}(\alpha), \boldsymbol{u}(\alpha) \right) \,, \qquad\qquad \alpha \in (0, \alpha_F] \,. \tag{29}$$

The initial condition for $\mathbb{Q}(0)$ is chosen to reproduce the random initialisation of the SGD algorithm. In particular, the initial first-layer weights and readout weights are drawn i.i.d. from a normal distribution with variances of $10^{-3}$ and $10^{-2}$, respectively. A thorough analysis of the validity of this ODE description is provided in Veiga et al. (2022), where the authors study the crossover between narrow and infinitely wide networks, clarifying the connection with the so-called *mean-field* or *hydrodynamic* regime (Mei et al., 2018; Chizat & Bach, 2018; Rotskoff & Vanden-Eijnden, 2022).

It is useful to write explicit expressions for the integrals involved in $f_{\mathbb{Q}}$ (Lee et al., 2021). First, expanding the terms in $\Delta^{(t)}$, we can write

$$f_{\boldsymbol{Q}, kh} = -\eta v_k^{(t_c)} \left[ \sum_{n=1}^{K} v_n^{(t_c)} I_3(n, k, h) - I_3(t_c, k, h) \right]$$
$$- \eta v_h^{(t_c)} \left[ \sum_{n=1}^{K} v_n^{(t_c)} I_3(n, h, k) - I_3(t_c, h, k) \right]$$
$$+ \eta^2 v_k^{(t_c)} v_h^{(t_c)} \left[ \sum_{n,m=1}^{K} v_n^{(t_c)} v_m^{(t_c)} I_4(n, m, k, h) + I_4(t_c, t_c, k, h) \right. \tag{30}$$
$$\left. - 2 \sum_{n=1}^{K} v_n^{(t_c)} I_4(n, t_c, k, h) \right]$$

$$f_{\boldsymbol{M}, kt} = -\eta v_k^{(t_c)} \sum_{n=1}^{K} v_n^{(t_c)} I_3(n, k, t) + \eta v_k^{(t_c)} I_3(t_c, k, t) \,, \tag{31}$$

$$f_{\boldsymbol{V}, tk} = \eta \left[ -\sum_{n=1}^{K} v_n^{(t_c)} I_2(k, n) + I_2(k, t_c) \right] \delta_{t, t_c} \,. \tag{32}$$

Similarly as in Eq. 19, we adopt the unified notation for the integrals

$$
\begin{aligned}
I_3(\beta, \rho, \zeta) &:= \mathbb{E}_{\boldsymbol{\lambda}, \boldsymbol{\lambda}_*} \left[ \lambda_\beta g'_\rho(\lambda_\rho) g(\lambda_\zeta) \right] \ , \\
I_4(\beta, \rho, \zeta, \tau) &:= \mathbb{E}_{\boldsymbol{\lambda}, \boldsymbol{\lambda}_*} \left[ g_\beta(\lambda_\beta) g_\rho(\lambda_\rho) g'_\zeta(\lambda_\zeta) g'_\tau(\lambda_\tau) \right] \ ,
\end{aligned}
\tag{33}
$$

where $\beta, \rho, \zeta, \tau$ can refer both to the indices of the student weights $k, h, n, m$ or the tasks $t, t_c$. In the special case $g(z) = g_*(z) = \mathrm{erf}(z/\sqrt{2})$, the integrals have explicit expressions as a function of the overlaps

$$
\begin{aligned}
I_3(\beta, \rho, \zeta) &= \frac{2 q_{\rho\zeta}(1 + q_{\beta\beta}) - 2 q_{\beta\rho} q_{\beta\zeta}}{\pi \sqrt{\Lambda_3}(1 + q_{\beta\beta})} \ , \\
I_4(\beta, \rho, \zeta, \tau) &= \frac{4}{\pi^2 \sqrt{\Lambda_4}} \arcsin \frac{\Lambda_0}{\sqrt{\Lambda_1 \Lambda_2}} \ ,
\end{aligned}
\tag{34}
$$

the symbol $q$ denotes generically an overlap from Eq. 16, and

$$
\begin{aligned}
\Lambda_0 &= \Lambda_4 \, q_{\beta\rho} - q_{\beta\tau} \, q_{\rho\tau} \, (1 + q_{\zeta\zeta}) - q_{\beta\zeta} \, q_{\rho\zeta} \, (1 + q_{\tau\tau}) + q_{\zeta\tau} \, q_{\beta\zeta} \, q_{\rho\tau} + q_{\zeta\tau} \, q_{\rho\zeta} \, q_{\beta\tau} \ , \\
\Lambda_1 &= \Lambda_4 \, (1 + q_{\beta\beta}) - q_{\beta\tau}^2 \, (1 + q_{\zeta\zeta}) - q_{\beta\zeta}^2 \, (1 + q_{\tau\tau}) + 2 q_{\zeta\tau} q_{\beta\zeta} \, q_{\beta\tau} \ , \\
\Lambda_2 &= \Lambda_4 \, (1 + q_{\rho\rho}) - q_{\rho\tau}^2 \, (1 + q_{\zeta\zeta}) - q_{\rho\zeta}^2 \, (1 + q_{\tau\tau}) + 2 q_{\zeta\tau} q_{\rho\zeta} q_{\rho\tau} \ , \\
\Lambda_3 &= (1 + q_{\beta\beta})(1 + q_{\rho\rho}) - q_{\beta\rho}^2 \ , \\
\Lambda_4 &= (1 + q_{\zeta\zeta}) \, (1 + q_{\tau\tau}) - q_{\zeta\tau}^2 \ .
\end{aligned}
\tag{35}
$$

## A.3 INFORMAL DERIVATION OF PONTRYAGIN MAXIMUM PRINCIPLE

Let us consider the augmented cost functional

$$
\mathcal{F}[\mathbb{Q}, \hat{\mathbb{Q}}, \boldsymbol{u}] = h\left(\mathbb{Q}(\alpha_F)\right) + \int_0^{\alpha_F} \mathrm{d}\alpha \ \hat{\mathbb{Q}}(\alpha)^\top \left[ -\frac{\mathrm{d}\mathbb{Q}(\alpha)}{\mathrm{d}\alpha} + f_{\mathbb{Q}}\left(\mathbb{Q}(\alpha), \boldsymbol{u}(\alpha)\right) \right] \ ,
\tag{36}
$$

where the conjugate variables $\hat{\mathbb{Q}}(\alpha)$ act as Lagrange multipliers, enforcing the dynamics at time $\alpha$. Setting to zero variations with respect to $\hat{\mathbb{Q}}(\alpha)$ results in the forward dynamics

$$
\frac{\delta \mathcal{F}[\mathbb{Q}, \hat{\mathbb{Q}}, \boldsymbol{u}]}{\delta \hat{\mathbb{Q}}(\alpha)} = 0 \Rightarrow \frac{\mathrm{d}\mathbb{Q}(\alpha)}{\mathrm{d}\alpha} = f_{\mathbb{Q}}\left(\mathbb{Q}(\alpha), \boldsymbol{u}(\alpha)\right) \ .
\tag{37}
$$

Integrating by parts, we find

$$
\begin{aligned}
\mathcal{F}[\mathbb{Q}, \hat{\mathbb{Q}}, \boldsymbol{u}] = h\left(\mathbb{Q}(\alpha_F)\right) &+ \int_0^{\alpha_F} \mathrm{d}\alpha \ \hat{\mathbb{Q}}(\alpha)^\top f_{\mathbb{Q}}\left(\mathbb{Q}(\alpha), \boldsymbol{u}(\alpha)\right) + \int_0^{\alpha_F} \mathrm{d}\alpha \frac{\mathrm{d}\hat{\mathbb{Q}}(\alpha)}{\mathrm{d}\alpha}^\top \mathbb{Q}(\alpha) \\
&- \hat{\mathbb{Q}}(\alpha_F)\mathbb{Q}(\alpha_F) + \hat{\mathbb{Q}}(0)\mathbb{Q}(0) \ .
\end{aligned}
\tag{38}
$$

Setting to zero variations with respect to $\mathbb{Q}(\alpha)$ for $0 < \alpha < \alpha_F$, we find the backward dynamics

$$
-\frac{\mathrm{d}\hat{\mathbb{Q}}(\alpha)^\top}{\mathrm{d}\alpha} = \hat{\mathbb{Q}}(\alpha)^\top \nabla_{\mathbb{Q}} f_{\mathbb{Q}}\left(\mathbb{Q}(\alpha), \boldsymbol{u}(\alpha)\right) \ ,
\tag{39}
$$

while for $\alpha = \alpha_F$ we get the final condition

$$
\hat{\mathbb{Q}}(\alpha_F) = \nabla_{\mathbb{Q}} h(\mathbb{Q}(\alpha_F)) \ .
\tag{40}
$$

Note that we do not consider variations with respect to $\mathbb{Q}(0)$ as this quantity is fixed by the initial condition $\mathbb{Q}(0) = \mathbb{Q}_0$. Finally, minimizing the cost functional with respect to the control $\boldsymbol{u}$, we get the optimality condition in Eq. 10 of the main text.

## A.4 OPTIMAL CONTROL FRAMEWORK

To determine the optimal control, we iterate Eqs. 5, 8, and 10 of the main text until convergence (Bechhoefer, 2021). Let us consider first the case where the control is the current task $t_c(\alpha)$, such that $t_c(\alpha) = t$ if the network is trained on task $t \in \{1, \ldots, T\}$ at training time $\alpha$. For simplicity, we

focus on the case $T = 2$, but the following discussion is easily generalised to any $T$. In particular, since here $u(\alpha) = t_c(\alpha)$ the evolution equation 5 can be written as

$$\frac{\mathrm{d}\mathbb{Q}(\alpha)}{\mathrm{d}\alpha} = f_\mathbb{Q}\left(\mathbb{Q}(\alpha), t_c(\alpha)\right), \quad \mathbb{Q}(0) = \mathbb{Q}_0. \tag{41}$$

Similarly, the backward dynamics reads

$$-\frac{\mathrm{d}\hat{\mathbb{Q}}(\alpha)^\top}{\mathrm{d}\alpha} = \hat{\mathbb{Q}}(\alpha)^\top \nabla_\mathbb{Q} f_\mathbb{Q}\left(\mathbb{Q}(\alpha), t_c(\alpha)\right), \tag{42}$$

with final condition

$$\hat{\mathbb{Q}}(\alpha_F) = \frac{1}{2}\nabla_\mathbb{Q}\varepsilon_1(\mathbb{Q}(\alpha_F)) + \frac{1}{2}\nabla_\mathbb{Q}\varepsilon_2(\mathbb{Q}(\alpha_F)). \tag{43}$$

The optimality equation 10 yields

$$t_c^*(\alpha) = \underset{t_c \in \{1,2\}}{\operatorname{argmin}}\left\{\hat{\mathbb{Q}}(\alpha)^\top f_\mathbb{Q}\left(\mathbb{Q}(\alpha), t_c(\alpha) = t_c\right)\right\}. \tag{44}$$

Therefore, we find the explicit formula for the optimal task protocol

$$t_c^*(\alpha) = \begin{cases} 1 & \text{if } \hat{\mathbb{Q}}(\alpha)^\top\left[f_\mathbb{Q}\left(\mathbb{Q}(\alpha), t_c(\alpha) = 2\right) - f_\mathbb{Q}\left(\mathbb{Q}(\alpha), t_c(\alpha) = 1\right)\right] > 0 \\ 2 & \text{otherwise.} \end{cases} \tag{45}$$

Then, we start from a guess for the control variable $t_c(\alpha)$. We integrate Eq. 41 forward, obtaining the trajectory $\mathbb{Q}(\alpha)$ for $\alpha \in (0, \alpha_F)$. Then, we integrate the backward equation 42, starting from the final condition 43, obtaining the trajectory $\hat{\mathbb{Q}}(\alpha)$ for $\alpha \in (0, \alpha_F)$. Then, the control variable can be updated using Eq. 45 and used in the next iteration of the algorithm. These equations 41, 42, and 45 are iterated until convergence.

We next consider the joint optimisation of the learning rate schedule $\eta(\alpha)$ and the task protocol $t_c(\alpha)$. The optimality condition 10 can be written as

$$(t_c^*(\alpha), \eta(\alpha)) = \underset{t_c \in \{1,2\}, \eta \in \mathbb{R}^+}{\operatorname{argmin}}\left\{\hat{\mathbb{Q}}(\alpha)^\top f_\mathbb{Q}\left(\mathbb{Q}(\alpha), (t_c(\alpha), \eta(\alpha)) = (t_c, \eta)\right)\right\}. \tag{46}$$

Crucially, the function $\hat{\mathbb{Q}}^\top f_\mathbb{Q}(\mathbb{Q}, (t_c, \eta))$ turns out to be quadratic in $\eta$. Explicitly,

$$\hat{\mathbb{Q}}^\top f_\mathbb{Q}(\mathbb{Q}, (t_c, \eta)) = a\eta^2 + b\eta, \tag{47}$$

where

$$a = \sum_{k,h=1}^K \hat{Q}_{kh} v_k^{(t_c)} v_h^{(t_c)}\left[\sum_{n,m=1}^K v_n^{(t_c)} v_m^{(t_c)} I_4(n, m, k, h) + I_4(t_c, t_c, k, h)\right. \tag{48}$$

$$\left. -2\sum_{n=1}^K v_n^{(t_c)} I_4(n, t_c, k, h)\right],$$

and

$$b = -\sum_{k,h=1}^K \hat{Q}_{kh}\left\{v_k^{(t_c)}\left[\sum_{n=1}^K v_n^{(t_c)} I_3(n, k, h) - I_3(t_c, k, h)\right]\right. \tag{49}$$

$$+ v_h^{(t_c)}\left[\sum_{n=1}^K v_n^{(t_c)} I_3(n, h, k) - I_3(t_c, h, k)\right]\right\}$$

$$- \sum_{k=1}^K \sum_{t=1}^T \hat{M}_{kt}\left[v_k^{(t_c)} \sum_{n=1}^K v_n^{(t_c)} I_3(n, k, t) - v_k^{(t_c)} I_3(t_c, k, t)\right]$$

$$+ \sum_{k=1}^K \hat{v}_k^{(t_c)}\left[-\sum_{n=1}^K v_n^{(t_c)} I_2(k, n) + I_2(k, t_c)\right].$$

Performing the minimization over $\eta$ first, we obtain

$$\eta^*(\alpha, t_c) = -\frac{b}{2a}.$$ (50)

The minimisation over $t_c$ yields

$$t_c^*(\alpha) = \begin{cases} 1 & \text{if } \hat{\mathbb{Q}}(\alpha)^\top \left[ f_\mathbb{Q}\left(\mathbb{Q}(\alpha), (1, \eta^*(\alpha, 1))\right) - f_\mathbb{Q}\left(\mathbb{Q}(\alpha), (2, \eta^*(\alpha, 2))\right) \right] > 0 \\ 2 & \text{otherwise.} \end{cases}$$ (51)

and hence

$$\eta^*(\alpha) = \eta^*(\alpha, t_c^*(\alpha)).$$ (52)

Interestingly, we observe that the learning rate schedule has a different functional form depending on the current task $t_c$. This can be seen in Fig. 4 where the learning rate switches between two different schedules depending on the current task $t_c$.

## B  READOUT LAYER CONVERGENCE PROPERTIES

In this appendix, we examine the asymptotic behaviour of the readout layer weights during the late stages of training. In particular, we are interested in the convergence rate as a function of the task similarity $\gamma$. As in the main text, we consider the case $K = T = 2$. From the overlap trajectories in Fig. 8 for $\gamma > 0.3$, we observe that the cosine similarity quickly approaches unity, i.e., $|M_{kt}|/\sqrt{Q_{kk}} \approx \delta_{kt}$, which corresponds to perfect feature recovery. Therefore, the decrease in performance for $\gamma > 0.3$ seen in Fig. 3 must be attributed to the dynamics of the second layer. Indeed, in Fig. 8, we observe a slowdown in the readout dynamics as $\gamma \to 1$.

Assuming perfect convergence of the feature layer to $\boldsymbol{w}_1 = \boldsymbol{w}_*^{(1)}$ and $\boldsymbol{w}_2 = \boldsymbol{w}_*^{(2)}$, we consider the dynamics of the readout layer while training on task $t = 1$. We expect the corresponding readout layer to converge to the specialised configuration $\boldsymbol{v}^{(1)} = (v_1^{(1)}, v_2^{(1)}) = (1, 0)^\top$ and we would like to compute the convergence rate as a function of $\gamma$. The dynamics of the readout layer reads

$$\frac{dv_1^{(1)}}{d\alpha} = \eta \left[ \frac{1}{3}(1 - v_1^{(1)}) - \frac{2}{\pi} \arcsin\left(\frac{\gamma}{2}\right) v_2^{(1)} \right],$$ (53)

$$\frac{dv_2^{(1)}}{d\alpha} = \eta \left[ \frac{2}{\pi} \arcsin\left(\frac{\gamma}{2}\right)(1 - v_1^{(1)}) - \frac{1}{3} v_2^{(1)} \right],$$

which can be rewritten as

$$\frac{d}{d\alpha} \begin{pmatrix} 1 - v_1^{(1)} \\ v_2^{(1)} \end{pmatrix} = \eta \boldsymbol{A} \begin{pmatrix} 1 - v_1^{(1)} \\ v_2^{(1)} \end{pmatrix},$$ (54)

where

$$\boldsymbol{A} = \begin{bmatrix} -1/3 & a \\ a & -1/3 \end{bmatrix},$$ (55)

and $a = 2 \arcsin(\gamma/2)/\pi$. Note that $a < 1/3$ for $0 < \gamma < 1$, hence $\boldsymbol{A}$ is negative definite, implying convergence to $\boldsymbol{v}^{(1)} = (1, 0)^\top$. The rate of convergence is determined by the smallest eigenvalue (in absolute value): $a - 1/3$. The associated convergence timescale is therefore

$$\alpha_{\text{conv}} = \frac{3\pi}{\eta(\pi - 6 \arcsin(\gamma/2))},$$ (56)

as anticipated in Eq. 11 of the main text.

## C  SUPPLEMENTARY FIGURES

### C.1  ADDITIONAL RESULTS IN THE SYNTHETIC FRAMEWORK

Fig. 8 describes the dynamics of the optimal replay strategy for different values of task similarity in the same setting as Fig. 3 of the main text. In particular, the upper panel displays the evolution of

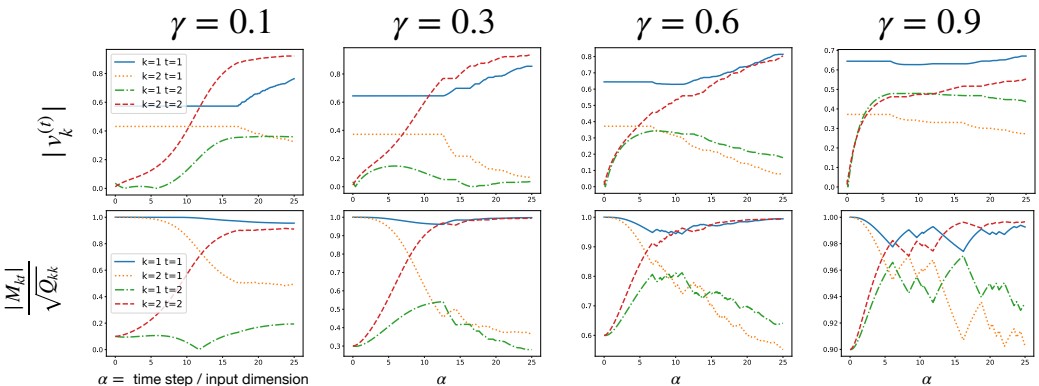

Figure 8: **Overlap dynamics with optimal replay.** We plot the absolute value of the task-dependent readout weights $|v_k^{(t)}|$ (upper panel) and the cosine similarity $|M_{kt}|/\sqrt{Q_{kk}}$ as a function of the training time $\alpha$. Different columns refer to different choices of task similarity $\gamma = 0.1, 0.3, 0.6, 0.9$.

the magnitude of the readout weights $|v_k^{(t)}|$, while the lower panel shows the trajectory of the cosine similarity $|M_{kt}|/\sqrt{Q_{kk}}$.

Fig. 9 compares the values of the loss at the end of training, averaged on both tasks, for different task-selection strategies. In particular, it highlights the performance gap between the four replay strategies at constant learning rate considered in the main text (no-replay, interleaved, optimal and pseudo-optimal) and the strategy that simultaneously optimise over task-selection and learning rate. Additionally, we consider exponential and power law learning rate schedules, in combination with interleaved replay protocol. For each value of task similarity $\gamma$, we optimize over the schedule parameters via grid search. We still find a performance gap with respect to the optimal strategy, which highlights the relevance of the joint optimization of training protocols.

Fig. 10 illustrates a continual learning setting with $T = 3$ tasks. The student is a two-layer neural networks with $K = 3$ hidden units, and a different readout is trained for each task. In the initial training phase, the student is trained on task 1 up to time $\alpha = 1000$, when the loss reaches $\sim 10^{-6}$. In the second phase, the student must learn tasks 2 and 3 without forgetting task 1. Panel **a)** shows the losses of the optimal strategy as a function of time for the three tasks and their average, during the second training phase. The optimal strategy is represented by a colour bar in the upper panel. It consists in an initial phase where only the new tasks 2 and 3 are presented to the network, and a second phase where task 1 is replayed. Despite its complicated structure, it shares some similarities with the pseudo-optimal strategy described in the main text. Specifically, task 1 is replayed only when its loss is comparable to the losses on the other two tasks. Panel **b)** shows the average loss as a function of time, comparing different task-selection protocols. In particular, we consider

- *interleaved* protocols, where two or all three tasks are alternated during training;
- *sequential* protocols, where tasks are presented in distinct blocks without being replayed;
- a *shuffled* protocol, that preserves the relative fraction of samples from each tasks obtained from the optimal strategy but presents them in a randomised order.

The final performance of the optimal strategy surpasses all the aforementioned approaches. Notably, as the number of tasks grows, the number of possible heuristic strategies expands significantly, making it difficult to identify effective solutions through intuition alone. This highlights the importance of a theoretical framework for systematically determining the optimal strategy.

## C.2 ADDITIONAL RESULTS ON REAL DATASET

We run additional experiments on real data using the simulation setup detailed in Sec. 3.2. We consider the CIFAR10 dataset Krizhevsky et al. (2009) and create two classification tasks taking as task 1 the classes with odd labels and the others as task 2. Fig. 11 shows the results of the training curve for the two tasks according to the three strategies and the final figure compares the average

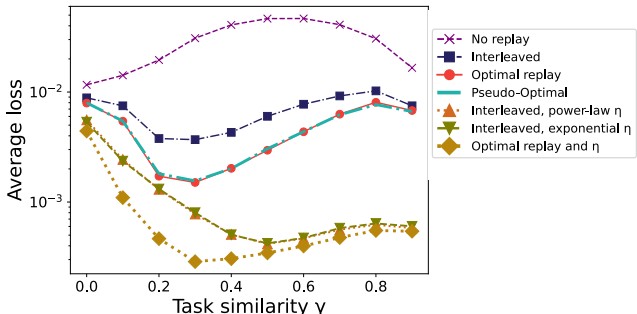

Figure 9: **Adopting an optimal learning rate schedule leads to major perfomance improvement.**
Average loss on both tasks at the end of the second training phase as a function of task similarity $\gamma$
under the same setting and parameters as Fig. 3 of the main text. The top four lines correspond to
different strategies at constant learning rate $\eta = 1$: no replay (purple crosses), optimal (red dots),
interleaved (blue squares), pseudo-optimal (cyan dashed line). The bottom three curves correspond
to different annealing schemes for the learning rate. Up-facing orange triangles correspond to inter-
leaved replay with power law annealing $\eta(\alpha) = \eta_0(1 - \alpha/\alpha_f)^\beta$, while down-facing green triangles
indicate exponential annealing $\eta(\alpha) = \eta_0 \exp(-\alpha/\alpha_0)$. For both annealing schedules, the scalar
parameters $\eta_0$, $\beta$, and $\alpha_0$ are optimized by grid search for each value of $\gamma$. Finally, the brown plus
signs correspond to jointly optimal replay and learning rate schedules (see Fig. 4).

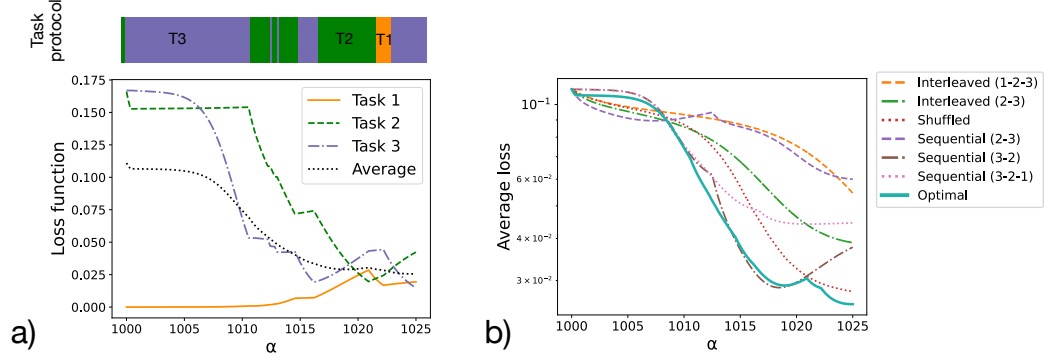

Figure 10: **Optimal replay schedule for $T = 3$ tasks.** The student network has $k = 3$ hidden units.
In the initial phase, $\alpha = [0, 1000]$ the network is trained on task 1 until convergence (loss is $10^{-6}$).
In the second phase $\alpha = [1000, 1025]$, we determine the optimal replay strategy. In both phases
the learning rate is constant $\eta = 1$. The tasks are chosen such that the overlaps are $S_{1,2} = 0.5$,
$S_{2,3} = 0.5$, and $S_{1,3} = 0$. Panel **a)** shows the optimal task protocol and the evolution of the loss
over the three tasks. The result is a complicated replay strategy, though it shares some similarities
with the pseudo-optimal strategy described in the main text. Specifically, task 1 is replayed only
when its loss is comparable to the losses on the other two tasks. Panel **b)** compares the optimal
strategy with different heuristics, all at $\eta = 1$. "Interleaved $(1 - 2 - 3)$" is an interleaved strategy
containing all tasks in equal proportion. "Interleaved $(2 - 3)$" is an interleaved strategy containing
tasks $(2 - 3)$ in equal proportion. "Shuffled" contains the same task proportions as in the optimal
strategy, but in random order (showing that the structure of the optimal replay strategy matters).
"Sequential $(2 - 3)$" corresponds to the sequential strategy with $t = 2$ in the first half and $t = 3$ in
the second. Similarly for "Sequential $(3 - 2)$". "Sequential $(3 - 2 - 1)$" has $t = 3$ in the first third
of the replay sequence, then $t = 2$ in the second third and $t = 1$ in the last third.

loss throughout the training steps. Notice that the observation reported in the controlled Fashion
MNIST experiment are still valid in this scenario.

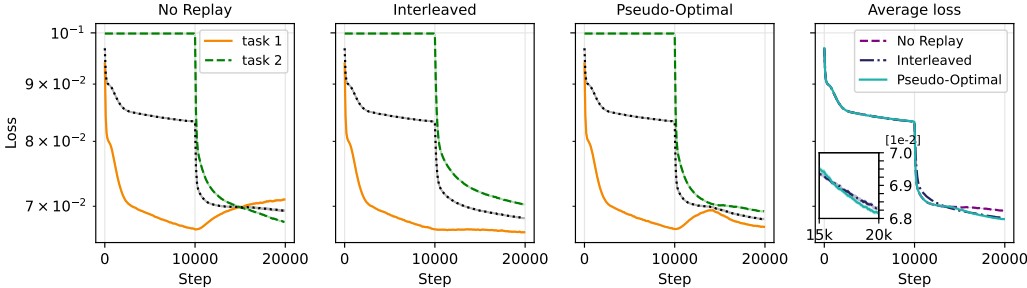

Figure 11: **Continual learning on CIFAR10.** Training curves of No-Replay, Interleaved, and Pseudo-Optimal learning strategies (from left to right) on CIFAR10. The two tasks are obtained partitioning the dataset according to the parity of the labels and training using online stochastic gradient descent. The final panel (rightmost) shows the average loss for the three strategies. The inset zooms on the final stage of learning.

As already observed in the main text, the performance of the Pseudo-Optimal learning strategies appears to interpolate between No-Replay and Interleaved. In Fig. 12 we highlight the differences between Pseudo-Optimal and the alternative strategies. We see a up to 3% improvement in performance during learning. Contrarily to the main text, we observe also a region where Pseudo-Optimal appears sub-performing with respect to the Interleaved strategies, however this is limited to a less than 0.5% decrease in loss.

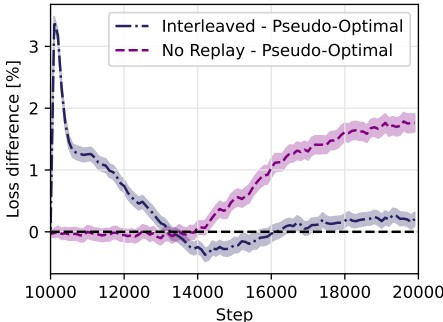

Figure 12: **Learning difference on CIFAR10.** This figure complement Fig. 11, highlighting the differences between the strategies.

