# OpenReview forum: "Optimal Protocols for Continual Learning via Statistical Physics and Control Theory"
_ICLR.cc/2025/Conference — ICLR 2025 Poster_

### Official Review · Reviewer_4ows · 2024-11-02

**Soundness:** 4
**Presentation:** 4
**Contribution:** 3
**Rating:** 8
**Confidence:** 4

**Summary:**

The authors present a theoretical framework based on a teacher-student setup to define optimal strategies for task switching and learning rate adjustments in continual learning settings. Their approach starts with a dimensionality reduction, which makes the problem more tractable by focusing on three types of overlaps between student and teacher instead of tracking each network weight. In the limit of infinite input dimension, this reduction effectively captures the high-dimensional stochastic dynamics of SGD through ODEs. The next step is to frame the problem of optimal task selection and learning rate scheduling as an optimal control problem, aiming to minimize the final generalization error across all tasks. They address this problem through a variational approach that iterates between forward dynamics (tracking SGD), backward dynamics (using conjugate parameters), and updating the control strategy (task selection/learning rate) until reaching an optimal protocol. Using synthetic experiments, they show that their optimal protocol has a structured design: an initial focus phase on the new task, followed by a revision phase with interleaved training, which performs best at intermediate task similarities. They validate their theoretical insights on Fashion-MNIST by constructing controlled task pairs through linear interpolation.

**Strengths:**

- The paper is well-written and organized, with a comprehensive discussion of related work.
- The topic addresses a significant gap in the theoretical understanding of continual learning. The insights presented are valuable for the continual learning community and likely of interest to a broader machine learning audience.
- The experiments are detailed, especially for a theoretical work.
- While the optimal theory relies on access to a 'teacher' model, which is not realistic in practical applications, the theoretical derivations remain insightful. Moreover, the authors introduce a 'pseudo-optimal' version that operates without direct access to this information, requiring only an estimate of generalization error on the tasks—something feasible in real-world scenarios.

**Weaknesses:**

- The theoretical framework assumes a "multi-headed" approach, where each task has its own output layer. However, most replay methods that interleave past tasks typically use a shared output layer. When using a shared head, the best approach is generally to mix all tasks in a batch, as tasks without associated samples get heavily penalized, pushing their output neurons towards inactivity -- neurons become dead. While the multi-headed setup make theory tractably—and is reasonable given the paper’s theoretical focus—this choice limits the immediate applicability of the findings to common practical implementations.

**Questions:**

- While the Fashion-MNIST interpolation setting enables direct control of task similarity and consistent with the theoretical framework, the paper would be stronger if it demonstrated results in settings that deviate somewhat from the theoretical conditions. For example, using MNIST with controlled pixel permutations would provide an alternative way to study task similarity. Such experiments would showcase the generalizability of the theoretical insights to different types of task relationships and data distributions. This additional experiment isn’t necessary, but some discussion in Section 3.2 on why this specific scenario was chosen would be helpful.
- The term 'epoch' in this paper may be misleading. While the authors use it to refer to training time scaled by input dimensions, 'epoch' conventionally means a single pass over the dataset in machine learning. This could cause confusion, especially in Section 3.2, where theoretical and practical results are compared. I suggest using a term like 'normalized training steps' to improve clarity.

---

> ### Author Response · Authors · 2024-11-19
> **Answer to Reviewer 4ows**
>
> We thank the reviewer for their valuable time reviewing our work and for their positive comments.
>
> > *The theoretical framework assumes a "multi-headed" approach, where each task has its own output layer. However, most replay methods that interleave past tasks typically use a shared output layer. When using a shared head, the best approach is generally to mix all tasks in a batch, as tasks without associated samples get heavily penalized, pushing their output neurons towards inactivity -- neurons become dead. While the multi-headed setup make theory tractably—and is reasonable given the paper’s theoretical focus—this choice limits the immediate applicability of the findings to common practical implementations.*
>
> We thank the reviewer for this insightful comment. The idea of increasing the number of classes while using a shared output layer and task-specific classes, all updated at each iteration, is indeed interesting and relevant. At present, the theoretical description of the dynamics, based on statistical physics, has not been extended to multi-label classification. However, we believe this extension is feasible. Once this preliminary step is accomplished, integrating the control theory component would follow naturally. We agree that this represents an interesting direction for future research and have now mentioned it in the revised manuscript. Thank you for highlighting this point.
>
> **Questions**
>
> > *While the Fashion-MNIST interpolation setting enables direct control of task similarity and consistent with the theoretical framework, the paper would be stronger if it demonstrated results in settings that deviate somewhat from the theoretical conditions. For example, using MNIST with controlled pixel permutations would provide an alternative way to study task similarity. Such experiments would showcase the generalizability of the theoretical insights to different types of task relationships and data distributions. This additional experiment isn’t necessary, but some discussion in Section 3.2 on why this specific scenario was chosen would be helpful.*
>
> We have welcomed the reviewer’s suggestion and run additional experiments on CIFAR10 without introducing any interpolation between tasks. We create two classification tasks taking as task 1 the classes with odd labels and the others as task 2. The results are presented in Appendix C.2 and confirm our observations on pseudo-optimal strategy striking the balance between no-replay and interleaved.
>
>
> > *The term 'epoch' in this paper may be misleading. While the authors use it to refer to training time scaled by input dimensions, 'epoch' conventionally means a single pass over the dataset in machine learning. This could cause confusion, especially in Section 3.2, where theoretical and practical results are compared. I suggest using a term like 'normalized training steps' to improve clarity.*
>
> We thank the reviewer for the suggestion, that we welcome to improve the clarity of our presentation. We have replaced the “training epoch” with “training step” everywhere in the manuscript.
>
>
> We hope our response addresses your concerns and welcome further suggestions to improve the clarity of our presentation. If you find our response convincing/useful, please consider increasing the score.

---

### Official Review · Reviewer_pAPx · 2024-11-03

**Soundness:** 3
**Presentation:** 4
**Contribution:** 4
**Rating:** 8
**Confidence:** 4

**Summary:**

The paper addresses the problem of catastrophic forgetting in neural networks during continual learning. The paper aims to find optimal task-selection protocols to minimize forgetting and maximize performance, moving beyond heuristic approaches. This is done through combining statistical physics with optimal control theory to derive closed-form formulas for training dynamics.

**Contributions:**
- Derive optimal task-selection protocols and learning rate schedules as functions of task similarity and problem parameters.
- Proposes a "pseudo-optimal" replay strategy with distinct focus and revision phases to minimize forgetting.

**Strengths:**

- Figures are clear, well explained, and easy to understand.
- The paper uses an original and refreshing approach with strong theoretical backing.
- Demonstrates robust alignment between theoretical predictions and experimental results.
- Complex mathematical formulations are well-supported.

**Weaknesses:**

- Please try to use shorter sentences. The sentences tend to be overly long and complex.
- The mathematics and equations are dense and could limit accessibility for readers unfamiliar with optimal control or statistical physics.
- The methodology might benefit from additional real-world datasets or further exploration in complex architectures for broader generalizability. Additional datasets with higher complexity (e.g., CIFAR-10/100 or domain-specific continual learning datasets) would better demonstrate the generalizability of the proposed protocols.
- Experiments primarily focus on two-task scenarios. Including more tasks would test the robustness and scalability of the approach, offering more insight into how it handles diverse, real-world continual learning settings.
- Although the paper suggests a posteriori interpretations, actionable guidelines on why specific phases are structured as they are (e.g., conditions where certain strategies excel) would improve applicability and user understanding.
- Theoretical findings rely on idealized assumptions (e.g., i.i.d. Gaussian inputs, simplified two-layer networks) which may not hold in complex, structured data environments. Expanding to other data models would strengthen claims of broader applicability.
- The model does not address relative task difficulty or task order impact, which are critical in real-world scenarios. Future extensions could explore adaptive protocols that consider task-specific attributes dynamically.

**Questions:**

- In many results, the pseudo-optimal method is comparable in performance to the naive interleaved. Could you better explain the advantages of using this pseudo-optimal method instead of a naive interleave?
- The "pseudo-optimal" strategy provides practical insights, but the underlying mechanics of the optimal task-selection structure could be further explained to make the protocol more interpretable.
- (see weaknesses for more suggestions)

---

> ### Author Response · Authors · 2024-11-19
> **Answer to Reviewer pAPx, pt. 1**
>
> We thank the reviewer for their valuable time reviewing our work and for their positive comments.
>
> > *Please try to use shorter sentences. The sentences tend to be overly long and complex.*
>
> We thank the reviewer for  this suggestion. We have improved the clarity of our presentation in the revised version of the manuscript.
>
> >*The mathematics and equations are dense and could limit accessibility for readers unfamiliar with optimal control or statistical physics.*
>
> We have revised Sec. 2 expanding the explanations and introducing additional pointers to previous works. We are happy to further improve our presentation if the reviewer has more specific suggestions in mind.
>
> > *The methodology might benefit from additional real-world datasets or further exploration in complex architectures for broader generalizability. Additional datasets with higher complexity (e.g., CIFAR-10/100 or domain-specific continual learning datasets) would better demonstrate the generalizability of the proposed protocols.*
>
> We have welcomed the reviewer’s suggestion and run additional experiments on CIFAR10, presented in Appendix C.2. These new results confirm our initial observations of Sec. 3.2. In particular, the performance of the pseudo-optimal learning strategy appears to interpolate between no-replay and interleaved depending on the dataset size.
>
> > *Experiments primarily focus on two-task scenarios. Including more tasks would test the robustness and scalability of the approach, offering more insight into how it handles diverse, real-world continual learning settings.*
>
> Our theoretical framework is directly applicable to more than two classes. We have now added a paragraph about the case T=3 in Appendix C.1, our results are shown in the additional Fig. 10. Also in this case, we find that the optimal strategy outperforms the heuristic benchmarks. The structure of the optimal strategies for T=3 is more complicated, but remarkably it still shares some similarities with the pseudo-optimal strategy described in the main text. Specifically, task 1 is replayed only when its loss is comparable to the losses on the other two tasks. Notably, as the number of tasks grows, the number of possible heuristic strategies expands significantly (we consider 6), making it difficult to identify effective solutions through intuition alone. This highlights the importance of a framework for systematically determining the optimal strategy. While the additional paragraph in Appendix C is mainly a proof of concept, leveraging insights from our theoretical framework to deal with multi-class continual learning in real-world settings is an exciting and feasible application of our theory that we leave for future work.
>
> > *Although the paper suggests a posteriori interpretations, actionable guidelines on why specific phases are structured as they are (e.g., conditions where certain strategies excel) would improve applicability and user understanding.*
>
> We thank the reviewer for this insightful suggestion. In response, we have expanded our explanation in Section 3.2. Specifically, we find that the size of the downstream task plays a crucial role. For small datasets, the no-replay strategy performs well, as the loss reduction on the new task outweighs the forgetting of previous tasks. Conversely, for large datasets, the interleaved strategy is more effective, as it mitigates forgetting while maintaining reasonable performance on the new task. The transition between these two phases depends on the specific dataset, as evidenced by the differences between Fig.6 (MNIST) and Fig.11 (CIFAR10). Identifying this transition point can be challenging in practice. However, the pseudo-optimal strategy we propose effectively interpolates between these two extremes, achieving near-optimal performance across both regimes.

---

> > ### Author Response · Authors · 2024-11-19
> > **Answer to Reviewer pAPx, pt. 2**
> >
> > > *Theoretical findings rely on idealized assumptions (e.g., i.i.d. Gaussian inputs, simplified two-layer networks) which may not hold in complex, structured data environments. Expanding to other data models would strengthen claims of broader applicability.*
> >
> > The reviewer correctly identifies this as a crucial extension of our theory. Structured data models have been extensively studied using statistical physics methods in recent years [1-5], providing a promising foundation for applying our optimal control framework. For instance, an interesting extension involving a more complex data model is the Hidden Manifold Model [3], where data is generated from a lower-dimensional space and embedded via a nonlinear mapping into a higher-dimensional space. The forward learning dynamics for this nonlinear model have been derived in [3], and we are confident that our optimal control approach can be directly applied to this setting.
> >
> > *References:*
> > [1] Mézard, Marc. Indian Journal of Physics (2023): 1-12. [2] Cagnetta, Francesco, et al. Physical Review X 14.3 (2024): 031001. [3] Goldt, Sebastian, et al. Physical Review X 10.4 (2020): 041044. [4] Loureiro, Bruno, et al. Advances in Neural Information Processing Systems 34 (2021): 18137-18151. [5] Wakhloo, Albert J., Tamara J. Sussman, and SueYeon Chung. Physical Review Letters 131.2 (2023): 027301.
> >
> > > *The model does not address relative task difficulty or task order impact, which are critical in real-world scenarios. Future extensions could explore adaptive protocols that consider task-specific attributes dynamically.*
> >
> > The reviewer raises an excellent point. Extending our model to address relative task difficulty and task order is a relevant direction we are actively exploring. Our framework offers several ways to incorporate task difficulty, such as by adjusting parameters of the input distribution or adding label noise. These adaptations can be directly incorporated in our theoretical framework and would allow us to develop adaptive protocols that dynamically respond to task difficulty and ordering. For instance, we discuss curriculum learning as a potential application in the conclusion.
> >
> >
> > **Questions**
> >
> > >*In many results, the pseudo-optimal method is comparable in performance to the naive interleaved. Could you better explain the advantages of using this pseudo-optimal method instead of a naive interleave?*
> >
> > The performance gap between pseudo-optimal and interleaved strategies depends on the size of the downstream dataset, or, equivalently, the duration of the second phase of training. In particular, the performance is comparable only for large enough downstream tasks. Notice that panel a) of figure 3 is for a fixed value of $\alpha_F=25$.
> > Fig. 7 shows more clearly how the optimal strategy depends on the number of available samples for the downstream task. For small downstream tasks, no-replay outperforms interleaved (second and third panels). The interleaved strategy gets closer to optimal as the size of the downstream dataset increases. The pseudo-optimal strategy, because of its structure, results in an interpolation between no-replay and interleaved strategies that automatically adapts to the dataset size, leading to the best performance in both regimes.
> >
> > This is precisely one of the messages that Fig. 7 aims to convey. We thank the reviewer for their concern and have now added additional explanations in the main text. We have confirmed this observation with additional experiments on CIFAR10, explained in Appendix C.2.
> >
> > >*The "pseudo-optimal" strategy provides practical insights, but the underlying mechanics of the optimal task-selection structure could be further explained to make the protocol more interpretable.*
> >
> > We would like to emphasize that standard meta-optimization methods, such as the one we employ, compute globally optimal curves based on model parameters, such as task similarity and dataset size. In principle, this could make the solution highly dependent on these parameters. However, we find it quite remarkable that our optimal strategy allows for an intuitive interpretation that is robust to the specifics of the theoretical model. Moreover, it can be explained solely in terms of the loss, enabling generalization to real datasets. In our view, this already represents a considerable level of interpretability.
> >
> > We hope our response addresses your concerns and welcome further suggestions to improve the clarity of our presentation. If you find our response convincing/useful, please consider increasing the score.

---

> > > ### Comment · Reviewer_pAPx · 2024-11-26
> > >
> > > Hello,
> > > Thank you for the highly detailed response and the corrections. This helps clarify the questions I had.

---

### Official Review · Reviewer_PcHB · 2024-11-04

**Soundness:** 3
**Presentation:** 3
**Contribution:** 3
**Rating:** 6
**Confidence:** 4

**Summary:**

This paper provides a theoretical framework for optimal task selection and step-size optimization in continual learning.
The authors use the student-teacher formalism where a task is determined by a linear teacher, and the student uses a neural network with 1 hidden layer.
Using linearized dynamics for the student's network, and ideas from optimal control, they identify an "optimal protocol" for sampling data from tasks, as well as for setting the step-size.
The ideas are thoroughly evaluated in a toy setting with simulated data, with some additional experiments on Fashion MNIST and MLPs.

**Strengths:**

- The optimal control perspective provided by the problem formulation and theoretical analysis is interesting. Using this optimal control perspective to inform continual learning seems novel. While I have questions and suggestions below, I found this section to be clear overall.

- The empirical analysis of the proposed method on the toy problem to be thorough and well-motivated.

**Weaknesses:**

The problem formulation, while interesting, seems ultimately misguided for most settings in which continual learning is desirable.

- The first problem is that, continual learning is motivated from the costly nature of retraining. With small/linear models, the costs (and challenges) of retraining the model are diminished. There does not seem to be an obvious extension for this theory to larger models, and so the applicability of this approach seems largely limited to linear/small models.

- The second is the "optimallity" of the proposed protocol is ultimately about speed of minimizing training error, and not the generalization quality of the model. On this front, the proposed approach could be generalized. Indeed, the results in Section 6 seem to report test loss, whereas most other plots report train loss.

**Questions:**

- Can you clarify how the multi-headed approach deals with training on previous tasks? Are the readout weights from task 1 updated with interleaved training on task 1?

- I understand that relevant details can be found in the appendix, but the presentation of the results in Section 2 can be improved. In particular, the overlaps that are introduced should be briefly described in how they dictate the optimization dynamics.

- The "dimensional reduction techiques from statistical physics" approach you describe, and the "overlaps" seem to be related to more familiar concepts in the deep learning theory literature. In particular, they seem analogous to the linearized dynamics described by the neural tangent kernel. Can you comment on any similarities or differences?

- "we consider K [hidden dim] = T [number of tasks] to guarantee that the student network has enough capacity to learn all tasks perfectly." I do not think this is necessarily the case. Is the additional assumption that the data is a scalar linear regression problem? If this is the case, this seems like a very limited setting.

- Section 3 and Figure 2: Why are the number of epochs in task 2 comparatively low? Is it the case that interleaved training can achieve perfect error on task 1 and task 2 asymptotically?

- Section 3.1 (agreement with theory): Can you comment on whether there are any differences between the assumptions made in the problem formulation and the experimental results in this section?

- Section 3 (impact of task similarity): Task similarity seems to be measured in terms of the absolute value of the cosine similarity, which makes sense for the linearity assumption. I wonder what can be said when this assumption does not hold, can any results be generalized to nonlinear data?

- Section 3 (Optimal learning rate schedules): It seems like your results were using vanilla SGD< how does the "optimal learning rate" compare to adaptive optimisers (e.g., momentum, gradient normalization, or hyperparameter-free optimisers)

- Section 3.2 (experiments on real data): You mention that training was on a squared loss, but this is a classification problem. How were the targets represented?


** Minor Comments
- I do not think the title accurately describes the contributions. First, the term optimal protocol is not common in the continual learning literature, and only defined towards the end of Section 1. Also, the connection to statistical physics is relegated to the Appendix.

---

> ### Author Response · Authors · 2024-11-19
> **Answer to Reviewer PcHB**
>
> We thank the reviewer for the precious feedback, which helped us to improve the communication of our findings.
>
> > *The first problem is that, continual learning is motivated from the costly nature of retraining. With small/linear models, the costs (and challenges) of retraining the model are diminished. There does not seem to be an obvious extension for this theory to larger models, and so the applicability of this approach seems largely limited to linear/small models.*
>
> First, we would like to clarify that our model is not linear. Both teachers have nonlinearities and a nonlinearity is also applied to the pre-activations of the student network. Our theory holds for arbitrary nonlinear activation functions. In the results section, we explicitly evaluate it using error functions for both teacher and student networks. Furthermore, we do not rely on linearization or any other approximations of the dynamics to obtain our analytical results. Crucially, our dynamical equations are *exact* in the high-dimensional limit. Regarding the applicability of our approach to larger models, we emphasize that it may not be essential to extend the full analytic derivation to state-of-the-art architectures and tasks in order to derive meaningful insights into optimal strategies. Instead, an effective approach would be to use interpretable strategies inferred from simpler models as a basis for training more complex, realistic models. The rationale is that certain hyperparameter schedules can be broadly effective and are not strongly dependent on model-specific details, much like widely used heuristics such as learning rate annealing. Furthermore, key extensions of our theory that would bring it closer to real-world data and models are within reach and are the focus of ongoing work. One promising example is the use of structured data models, which would allow us to incorporate some structure of real tasks directly into the theoretical framework. Structured data models have been extensively studied using statistical physics methods in recent years, and these recent advancements offer a promising foundation for applying our optimal control framework.
>
> *References:*
>
> - Mézard, Marc. Indian Journal of Physics (2023): 1-12.
> - Cagnetta, Francesco, et al. Physical Review X 14.3 (2024): 031001.
> - Goldt, Sebastian, et al. Physical Review X 10.4 (2020): 041044.
> - Loureiro, Bruno, et al. Advances in Neural Information Processing Systems 34 (2021): 18137-18151.
> - Wakhloo, Albert J., Tamara J. Sussman, and SueYeon Chung. Physical Review Letters 131.2 (2023): 027301.
>
> > *The second is the "optimallity" of the proposed protocol is ultimately about speed of minimizing training error, and not the generalization quality of the model. On this front, the proposed approach could be generalized. Indeed, the results in Section 6 seem to report test loss, whereas most other plots report train loss.*
>
> We would like to clarify that in the online learning setting there is no distinction between train and test loss, as each example is used only once. Consequently, there is no concept of a “loss landscape”, that is instead induced by the presence of a training dataset. For this reason, the online setting does not permit the study of the generalization gap, but this limitation is inherent to online learning itself and not specific to our application. Nonetheless, the online learning regime remains highly relevant to understand learning dynamics and continues to be widely studied in machine learning theory, see example references below. In the discussion section, we highlight extending our framework to batch learning as an exciting future direction, which would enable us to study the role of memorization, among other aspects.
>
> *References:*
>
> - Gerard Ben Arous, Reza Gheissari, and Aukosh Jagannath. The Journal of Machine Learning Research, 22(1):4788–4838, 2021.
> - Goldt, Sebastian, et al. Advances in neural information processing systems 32 (2019).
> - Song Mei, Andrea Montanari, and Phan-Minh Nguyen. Proceedings of the National Academy of Sciences, 115(33):E7665–E7671, 2018.
> - Lenaic Chizat and Francis Bach. Advances in neural information processing systems, 31, 2018.
> - Grant Rotskoff and Eric Vanden-Eijnden. Communications on Pure and Applied Mathematics, 75(9):1889–1935, 2022.
>
> We address the reviewer's questions in a separate comment. We hope our response addresses your concerns and welcome further suggestions to improve the clarity of our presentation. If you find our response convincing/useful, please consider increasing the score.

---

> > ### Author Response · Authors · 2024-11-19
> > **Answer to Reviewer PcHB's questions, pt.1**
> >
> > > *Can you clarify how the multi-headed approach deals with training on previous tasks? Are the readout weights from task 1 updated with interleaved training on task 1?*
> >
> > We consistently update the head corresponding to the task being presented: we update head 1 when task 1 is shown and head 2 when task 2 is shown. This setup aligns with the approach used by Lee et al. (2021, 2022). This is explicitly represented via the delta function in Eq. (13) of the appendix, and we have now added a clarifying remark.
> >
> > > *I understand that relevant details can be found in the appendix, but the presentation of the results in Section 2 can be improved. In particular, the overlaps that are introduced should be briefly described in how they dictate the optimization dynamics.*
> >
> > We have improved the presentation of Sec. 2, expanding the explanation of the overlap dynamics and introducing additional pointers to previous works. We are happy to further improve our presentation if the reviewer has more specific suggestions in mind.
> >
> > > *The "dimensional reduction techiques from statistical physics" approach you describe, and the "overlaps" seem to be related to more familiar concepts in the deep learning theory literature. In particular, they seem analogous to the linearized dynamics described by the neural tangent kernel. Can you comment on any similarities or differences?*
> >
> > Thank you for raising this point, it is indeed an important distinction. We emphasize that our theory is not analogous to linearized dynamics as described by the neural tangent kernel (NTK). Our framework is fundamentally different from the NTK regime because we operate entirely within the feature-learning regime, where both representations and readouts are continuously updated to learn a non-linear mapping.
> >
> > In the NTK framework, the number of hidden units must diverge according to a specific scaling, effectively linearizing the dynamics and restricting the network’s capacity for feature learning. In contrast, our approach assumes a high-dimensional input space, such as in image classification tasks where dimensions like pixels, colors, and channels naturally yield a high-dimensional structure (e.g., CIFAR has a dimensionality of ~3000, which is sufficient for our purposes). This high-dimensional input assumption supports the non-linear, feature-learning dynamics that distinguish our approach from the NTK regime.
> >
> > > *"we consider K [hidden dim] = T [number of tasks] to guarantee that the student network has enough capacity to learn all tasks perfectly." I do not think this is necessarily the case. Is the additional assumption that the data is a scalar linear regression problem? If this is the case, this seems like a very limited setting.*
> >
> > There is no additional assumption here and the problem is not linear. Indeed, the tasks are correlated in a nonlinear way by the mapping from input to hidden layer which involves a nonlinear activation function. For the student network to be able, in principle, to learn the teacher functions perfectly, K must be at least as large as T, regardless of the specifics of the input data and the teacher activations, even in the ideal scenario where student and teacher activations are aligned. We now show this explicitly in Appendix A. Let us stress that our equations are valid for arbitrary values of K and T, as long as they remain finite while the input dimension diverges. In this work, we focus on the setup of Lee, et al. (2021,2022) that assumes single-layer teacher networks and K = T. This choice makes our result directly comparable with existing literature.
> >
> > >*Section 3 and Figure 2: Why are the number of epochs in task 2 comparatively low? Is it the case that interleaved training can achieve perfect error on task 1 and task 2 asymptotically?*
> >
> > We assume the classifier is extensively trained on task 1 before exposure to task 2. This setup is particularly relevant for studying catastrophic forgetting, as it tests how well task 1 is retained after near-perfect performance. However, our framework is not restricted to this scenario. For instance, Figure 5 illustrates the opposite case, where both tasks are learned from scratch within a fixed number of epochs. Intermediate scenarios can also be explored by adjusting the durations of the first and second training phases, which are parameters in our theory. In this simple setting, in the absence of label noise, the error is expected to approach zero given an infinite amount of data, much larger than the model size. However, we focus on the more interesting regime for studying forgetting, where the dataset size is comparable to the number of model parameters.

---

> > > ### Author Response · Authors · 2024-11-19
> > > **Answer to Reviewer PcHB's questions, pt.2**
> > >
> > > > *Section 3.1 (agreement with theory): Can you comment on whether there are any differences between the assumptions made in the problem formulation and the experimental results in this section?*
> > >
> > > The main difference between theory and numerical simulations in Figure 2 is that the theory is formulated in the infinite-dimensional limit, whereas the simulations are performed at finite input dimension. Additionally, we present results from a single simulation instance rather than an average across instances to highlight the remarkable agreement, as the overlaps concentrate around their average values, as described by the equations in Appendix A.2
> > >
> > > > *Section 3 (impact of task similarity): Task similarity seems to be measured in terms of the absolute value of the cosine similarity, which makes sense for the linearity assumption. I wonder what can be said when this assumption does not hold, can any results be generalized to nonlinear data?*
> > >
> > > We are not sure what the reviewer means by “linearity assumption”. The main assumption on the input distribution in our theoretical model is Gaussianity. Our framework could naturally be extended to a Gaussian mixture model as the input distribution. An interesting extension of our results, involving a more complex input distribution, is to consider the Hidden Manifold Model [1]. In this case, the data is generated from a lower-dimensional space and embedded through a nonlinear mapping into a higher-dimensional space. The forward learning dynamics for this nonlinear data model have been derived in [1], and we are confident that our optimal control approach applies directly to this setting.
> > >
> > > *Reference:*
> > >
> > > [1] Goldt, Sebastian, et al. ``Modeling the influence of data structure on learning in neural networks: The hidden manifold model''. Physical Review X 10.4 (2020): 041044.
> > >
> > >
> > > > *Section 3 (Optimal learning rate schedules): It seems like your results were using vanilla SGD< how does the "optimal learning rate" compare to adaptive optimisers (e.g., momentum, gradient normalization, or hyperparameter-free optimisers)*
> > >
> > > To the best of our knowledge, the theoretical framework used to analyze forward learning dynamics in the online setting under consideration has not yet been extended to adaptive optimizers. While we agree that investigating adaptive optimization methods would be highly valuable, such an extension would require a substantial technical contribution in its own right.
> > >
> > > We have run additional benchmarks for the optimal learning rate, including exponential and power-law schedules (Appendix C.1).
> > >
> > > > *Section 3.2 (experiments on real data): You mention that training was on a squared loss, but this is a classification problem. How were the targets represented?*
> > >
> > > Like most theoretical studies on dynamics, our approach adopts the square loss for its mathematical tractability. In Sec. 3.2, we train using the square loss with binary labels to maintain consistency with the theoretical framework.
> > >
> > > >*Minor Comments: I do not think the title accurately describes the contributions. First, the term optimal protocol is not common in the continual learning literature, and only defined towards the end of Section 1. Also, the connection to statistical physics is relegated to the Appendix.*
> > >
> > > Thank you for your feedback on the title. We recognize that 'optimal protocol' may be less common in the continual learning literature, but we believe it effectively captures the core contribution of our work, as our primary focus is on identifying strategies that optimize learning performance. Similarly, while the full analytic characterization based on statistical physics appears in the Appendix, it forms the theoretical foundation underlying our approach.

---

> ### Comment · Reviewer_PcHB · 2024-11-26
>
> I thank the authors for their detailed reply, you have cleared up many questions. I want to like this paper, the optimal control approach is very interesting. I think my main issue is still the first point, on which I would appreciate further discussion:
>
> - Context: Continual learning is primarily about dealing with the trade-off of stability and plasticity. Catastrophic forgetting often occurs, meaning that performance on an old task can suffer while training on a new task. This is indeed a problem for which many have devised methods to solve. In particular, these approaches try to avoid logging experience from past tasks. However, if the experience from the previous task is logged then training can be interleaved, as your paper uses as a baseline.
>
> - The problem being studied here is, for the most part, task selection. My primary concern is that it is not clear if the task selection problem is an existing problem that is of interest. The authors themselves note in the related works: "to the best of our knowledge, the problem of optimal task selection has not been explored yet." I can see certain problems for which this can matter, such as in curriculum learning, particularly in reinforcement learning. But this literature may have differences in the specific problem formulation, as well as existing strong baselines.
>
> - Reprieve: I understand that your theoretical contributions are novel and interesting, but I think the underlying problem addressed by theoretical considerations should also be interesting.
>   Step-size adaptation can be potentially much more influential in learning, but this would require comparison with the many adaptive and hyperparameter free optimisers.
>
> (Edit: I want to be clear that I am not looking for additional results here. Rather, I am looking for more discussion and justification on the choice of problem being studied.)
>
>
> I also have some further clarifications regarding some of your answers to my questions:
>
> - On your high-dimensional limit vs NTK: To clarify, your "overlaps" involve a limit of the input dimension and the NTK involves a limit in the width. Is this the primary difference? This is worth highlighting, and discussing in relation to NTK so as to familiarize readers.
>
> - On (non)linearity of student and teacher: yes, I understand that there is a nonlinear activation function (even for the teacher), but this is basically a generalized linear family which seems necessarily limited compared to deeper hierarchical models. Is there a universal approximation theorem in the limit of large input dimension on a generalized linear function? (Not that universal approximation of the teacher-family is required for your problem of study to be interesting, but it would be a limitation.)
>
> - On test vs train in online setting: I did see that you consider the online setting, but some of your figures reference "epoch" which would involve training on the same dataset more than once. You should clarify what epoch means in this case, and perhaps be more explicit.
>
> - Your references in the rebuttal (pt1): listing papers without any context does little to help connect these paper with any particular claims you are making in the rebuttal.
>
> - About task similarity and linearity: Correlation and cosine similarity are inherently linear relationships. That is, there exist nonlinear relationships which can exhibit 0 correlation. My question was directed at whether this linearity assumption (of task similarity) is a limitation.

---

> > ### Author Response · Authors · 2024-11-27
> > **Clarification on the first point, pt. 1**
> >
> > Thank you for your feedback and for highlighting the value of our approach. Let us clarify your main concern further.
> >
> > >_Context: Continual learning is primarily about dealing with the trade-off of stability and plasticity. Catastrophic forgetting often occurs, meaning that performance on an old task can suffer while training on a new task. This is indeed a problem for which many have devised methods to solve. In particular, these approaches try to avoid logging experience from past tasks. However, if the experience from the previous task is logged then training can be interleaved, as your paper uses as a baseline._
> >
> > Replay-based approaches are well-established and successful mitigation strategies to prevent catastrophic forgetting (see, e.g., Sec. 4.2 of this recent review [1]). Therefore, a precise theoretical understanding of their optimality is a timely open question with practical implications. Our theory offers theoretically grounded answers to precisely these questions: _When is replay optimal to mitigate forgetting? When can it be avoided?_
> >
> > Our paper shows that the answer can depend on: (1) the available compute time, (2) the similarity between tasks.
> > Specifically, our findings suggest that when compute resources are low or tasks are highly dissimilar, replay can be avoided without sacrificing performance.
> > Notice that we always fix the total compute time ($\alpha_F$), so replay is not artificially advantaged—logging past experiences necessarily reduces time spent on new tasks.
> >
> > While this paper focuses on replay-based methods, that still lack fundamental understanding as shown throughout the paper, our approach can be extended to analyze other prominent strategies discussed in Sec. 4 of [1]. For example, it would be particularly interesting to explore: (1) regularization-based methods that mitigate forgetting; (2) dynamic pruning techniques that prevent the loss of plasticity. These possible extensions highlight the broader relevance of our theoretical framework, offering promising directions for future research and demonstrating its impact beyond the scope of this specific work.
> >
> > Finally, a theoretical understanding of replay in continual  learning draws intriguing parallels with replay mechanisms in neuroscience. Indeed it has been theoreticized by [2] that the mammalian brain tackles similar challenges through two complementary systems: the fast-learning hippocampus and the slower-learning neocortex. These systems work together to consolidate memories, with replay serving as a crucial mechanism for transferring information from the hippocampus to the neocortex.
> >
> > _References_:
> >
> > [1] Wang et al.,  https://arxiv.org/pdf/2302.00487
> >
> > [2] McClelland, McNaughton, and O'Reilly (1995)
> >
> > > _The problem being studied here is, for the most part, task selection. My primary concern is that it is not clear if the task selection problem is an existing problem that is of interest. The authors themselves note in the related works: "to the best of our knowledge, the problem of optimal task selection has not been explored yet." I can see certain problems for which this can matter, such as in curriculum learning, particularly in reinforcement learning. But this literature may have differences in the specific problem formulation, as well as existing strong baselines._
> >
> > We would like to clarify that the primary focus of our paper is **replay** as a strategy to mitigate forgetting in continual learning. To emphasize the broad applicability of our theory, we frame this problem within the more general context of “task selection optimization.”
> >
> > Maybe our sentence “to the best of our knowledge, the problem of **optimal** task selection has not been explored yet”, was not clear. The impact of task ordering has indeed been studied in applications (see, e.g., https://arxiv.org/pdf/2205.13323 and references therein). However, our contribution lies in investigating its _optimality_ using a principled theoretical framework, as opposed to heuristic approaches. Specifically, the optimal control approach we propose is novel. Replay per se is one of the most successful strategies to address catastrophic forgetting in applications.
> > Furthermore, tasks do not always come with an inherent notion of difficulty, which is a key distinction between our framework and curriculum learning. However, task selection, as a broader framework, encompasses problems like curriculum learning, that are indeed the object of ongoing work.

---

> ### Author Response · Authors · 2024-11-27
> **Clarification on the first point, pt. 2**
>
> > _Reprieve: I understand that your theoretical contributions are novel and interesting, but I think the underlying problem addressed by theoretical considerations should also be interesting. Step-size adaptation can be potentially much more influential in learning, but this would require comparison with the many adaptive and hyperparameter free optimisers._
>
> We hope to have clarified that our theory addresses the core problem of determining the **optimality of replay-based approaches** to mitigate catastrophic forgetting. While replay is a well-established technique for tackling the broader challenge of continual learning, current methods are still predominantly heuristic. Our work moves beyond these heuristics to provide a principled framework.
>
> The paragraph on “Optimal learning rate schedules” (Sec. 3.1) is precisely addressing the question: _What is the optimal adaptive learning rate for this problem?_ What is interesting is that: (1) we can find the optimal learning rate dynamics without resorting to heuristic methods; (2) this is achieved in conjunction with the optimization of the replay schedule.
> If the reviewer is suggesting that adaptive step-size methods alone could resolve catastrophic forgetting, we respectfully disagree. This phenomenon persists even in applications where such methods are widely used. Anyways, our theory already provides the optimal learning rate schedule, and removing the optimization over replay can only degrade performance.

---

> > ### Author Response · Authors · 2024-11-27
> > **Further clarifications, pt. 1**
> >
> > >_On your high-dimensional limit vs NTK: To clarify, your "overlaps" involve a limit of the input dimension and the NTK involves a limit in the width. Is this the primary difference? This is worth highlighting, and discussing in relation to NTK so as to familiarize readers._
> >
> > Yes, to simplify, the limit that we consider in this paper is in the input while the NTK assumes diverging hidden layers. There are some additional considerations about the right scaling of the layers that will guarantee the quantities to concentrate in the limit. A thorough analysis of the validity of our ODE description under various conditions has been conducted in [Veiga et al., Advances in Neural Information Processing Systems 35 (2022): 23244-23255]. Specifically, the authors investigate the crossover between narrow and infinitely wide networks, in conjunction with different learning rate scalings. We have included a comment on this point in Appendix A.2.
> >
> >
> > >_On (non)linearity of student and teacher: yes, I understand that there is a nonlinear activation function (even for the teacher), but this is basically a generalized linear family which seems necessarily limited compared to deeper hierarchical models. Is there a universal approximation theorem in the limit of large input dimension on a generalized linear function? (Not that universal approximation of the teacher-family is required for your problem of study to be interesting, but it would be a limitation.)_
> >
> > Unfortunately, theoretical frameworks for the analysis of neural networks inevitably have their limitations. As the reviewer points out, a narrow network as the one we study is not a universal approximator. However, it is expressive enough to represent the task under consideration.
> >
> > To the best of our knowledge, the analytic characterization of the dynamics of feature learning in deep nonlinear networks remains an open challenge. We would appreciate clarification on what the reviewer specifically means by the theory of “deeper hierarchical models”, as this could help us address their concerns more precisely.
> >
> >
> > >_On test vs train in online setting: I did see that you consider the online setting, but some of your figures reference "epoch" which would involve training on the same dataset more than once. You should clarify what epoch means in this case, and perhaps be more explicit._
> >
> > We kindly ask the reviewer to check the updated version of the paper where, in response to a concern of Reviewer 4ows, we substituted “epoch” with “step” in the text and figures. We hope this clarifies.
> >
> > >_About task similarity and linearity: Correlation and cosine similarity are inherently linear relationships. That is, there exist nonlinear relationships which can exhibit 0 correlation. My question was directed at whether this linearity assumption (of task similarity) is a limitation._
> >
> > We see the reviewer's concern, thank you for clarifying. In our problem, the cosine similarity is the right measure despite the non-linearity. This is a non-trivial observation initially reported by [Saad & Solla 1995, Biehl & Schwarze 1995] and later proved by [Goldt et al. 2019]. Indeed we can rigorously show that despite the non-linear nature of the problem the sufficient statistics that characterise the dynamics and the performance are given by the cosine similarities. We will add this comment to the revised version.

---

> > > ### Author Response · Authors · 2024-11-27
> > > **Further clarifications (References), pt. 2**
> > >
> > > >_Your references in the rebuttal (pt1): listing papers without any context does little to help connect these paper with any particular claims you are making in the rebuttal._
> > >
> > > The references in the first answer refer to various models of data structure that have been recently proposed and could be incorporated into our theoretical framework. The references in the second answer were intended to show the relevance of online learning in the recent machine learning literature.
> > >
> > > We apologize for any confusion, below we provide additional context to the references listed.
> > >
> > >
> > > **Structured data models**
> > >
> > > **[1]** _Mézard, Marc. Indian Journal of Physics (2023): 1-12._
> > >
> > > This is a review of the most influential techniques in the theory of spin glasses and how these can be applied to problems arising in inference and machine learning. It discusses the general open challenge of devising structured data models to study deep neural networks.
> > >
> > > **[2]** _Cagnetta, Francesco, et al. Physical Review X 14.3 (2024): 031001._
> > >
> > > The authors propose the “Random Hierarchy Model” to mimic the hierarchical structure of language and images, where classes are defined by groups of high-level features composed of interchangeable subfeatures following a hierarchy.
> > >
> > > **[3]** _Goldt, Sebastian, et al. Physical Review X 10.4 (2020): 041044._
> > >
> > > The work introduces the “Hidden Manifold Model”, where the data is generated from a lower-dimensional space and embedded through a nonlinear mapping into a higher-dimensional space.
> > >
> > > **[4]** _Loureiro, Bruno, et al. Advances in Neural Information Processing Systems 34 (2021): 18137-18151._
> > >
> > > The work proposes a generalized teacher-student model where the teacher and student can act on different spaces, generated with fixed, but generic feature maps to capture the behaviour of realistic data sets.
> > >
> > > **[5]** _Wakhloo, Albert J., Tamara J. Sussman, and SueYeon Chung. Physical Review Letters 131.2 (2023): 027301._
> > >
> > > The authors study a structured model of “object manifolds”, with arbitrary correlations between object representations, to account for representation invariances of the same object.
> > >
> > > **Online learning**
> > >
> > > **[1]** _Ben Arous, et al. The Journal of Machine Learning Research, 22(1):4788–4838, 2021._
> > >
> > > The authors proved the validity of the SGD limit in linear regression with Gaussian covariates using results from stochastic process and probability theory. Furthermore, the same author showed in a follow-up paper the effect of overparameterization in relation to the lottery ticket hypothesis. Additionally, the same framework was used to show the advantage of 2-layer neural networks with respect to kernel methods.
> > >
> > > **[2]** _Goldt, et al. Advances in neural information processing systems 32 (2019)._
> > >
> > > This work analogously studied the effect of overparameterization, showing the emergence of a complex dynamics characterised by the presence of attractive sub-optimal fixed points.
> > >
> > > The works **[3-5]** below independently showed the emergence of another interesting limit for the analysis of infinitely wide neural networks that takes many names: mean-field limit, hydrodynamic limit, and birth-death dynamics. The result showed that under the Wasserstein distance, the learning landscape appears convex.
> > >
> > >
> > > **[3]** _Mei, et al. Proceedings of the National Academy of Sciences, 115(33):E7665–E7671, 2018._
> > >
> > > **[4]** _Chizat, and Bach. Advances in neural information processing systems, 31, 2018._
> > >
> > > **[5]** _Rotskoff, and Vanden-Eijnden. Communications on Pure and Applied Mathematics, 75(9):1889–1935, 2022._

---

> > > > ### Comment · Reviewer_PcHB · 2024-12-02
> > > >
> > > > I would like to thank the authors for their in-depth response to my questions. I still think the problem of optimal sampling to combat forgetting is the primary limitation. However, this paper meets the bar for acceptance and I will update my score (5->6) to reflect this.

---

### Official Review · Reviewer_vgxp · 2024-11-06

**Soundness:** 3
**Presentation:** 3
**Contribution:** 3
**Rating:** 6
**Confidence:** 3

**Summary:**

The paper proposes an ODE relating network parameters, training control parameters (such as learning rate and which task to learn on), and the final performance of the model. Using this, it derives optimal protocols for the training control parameters (i.e. at each step, which task to train on and what learning rate to use) during continual learning. They notice that the task protocol can be approximated by a heuristic where the new task is trained on by the network until its loss matches the loss of the previous task, and then data from the two tasks is interleaved. They also derive an optimal learning rate schedule. They test their protocols on a synthetic task of matching output generated by two randomly initialized neural networks and a version of Fashion MNIST where each task is composed of picking 2 classes from the dataset. On the synthetic tasks, their protocols significantly outperform previously proposed protocols, while the improvements are more modest on Fashion MNIST.

**Strengths:**

- The idea itself is very interesting, and I liked the fact that the theoretically optimal protocol can be approximated by a simple heuristic. If the results transfer, the “pseudo-optimal” protocol could be very useful.
- At least on the synthetic tasks, the replay protocol seems to outperform the standard interleaved protocol across essentially all settings.

**Weaknesses:**

- The learning rate experiments need more experimental support to be convincing. The learned schedule is being compared to a constant learning rate schedule. It should also be compared to more standard learning rate schedules.
- It’s unclear how much these results can transfer to harder/“real world” settings. See question 1 below. On the one “real world” task, there does not seem to be any noticeable difference between the interleaved and pseudo-optimal strategy (Figure 7).
- Section 2 explaining the machinery behind finding the optimal control parameters can be explained better. It is a bit difficult to follow for someone not already familiar with the topic, which many people who are attempting to use the results of this paper (CL researchers) would likely not be.

**Questions:**

- It would be nice to see if some of these results transfer to harder continual learning datasets. Specifically, these results are all done on two class classification tasks. Do they hold with more classes? Also do they hold for non MLP networks? Do they hold for datasets such as split Imagenet or split CIFAR? Do they hold across more than 2 tasks?

---

> ### Author Response · Authors · 2024-11-19
> **Response to Reviewer vgxp**
>
> We thank the reviewer for the precious feedback, which helped us to improve the communication of our findings.
>
> > *The learning rate experiments need more experimental support to be convincing. The learned schedule is being compared to a constant learning rate schedule. It should also be compared to more standard learning rate schedules.*
>
> We have run additional experiments on the effect of learning rate annealing, shown in the updated Fig. 9 of Appendix C.1. In particular, we consider exponential and power law learning rate schedules, in combination with interleaved replay protocol. For each value of task similarity, we optimize over the schedule parameters via grid search. We still find a performance gap with respect to the optimal strategy, which highlights the relevance of the joint optimization of training protocols.
>
> > *It’s unclear how much these results can transfer to harder/“real world” settings. See question 1 below. On the one “real world” task, there does not seem to be any noticeable difference between the interleaved and pseudo-optimal strategy (Figure 7).*
>
> The optimal strategy depends on the number of available samples for the downstream task. For small downstream tasks, no-replay outperforms interleaved (second and third panels). The interleaved strategy gets closer to optimal as the size of the downstream dataset increases. The pseudo-optimal strategy, because of its structure, results in an interpolation between no-replay and interleaved strategies that automatically adapts to the dataset size, leading to the best performance in both regimes.
>
> We have added a new experiment on CIFAR10 that confirms this observation. This result is reported and explained in Appendix C.2.
>
> This is precisely one of the messages that Fig. 7 aims to convey. We thank the reviewer for reporting their concern, this led us to include additional explanations improving clarity in the main text. We hope that now it is clear that, from left to right in the plots, the performance curves of the interleaved and no-replay strategies gradually shift relative to each other, with the first becoming closer to optimal as the other moves further away.
>
> > *Section 2 explaining the machinery behind finding the optimal control parameters can be explained better. It is a bit difficult to follow for someone not already familiar with the topic, which many people who are attempting to use the results of this paper (CL researchers) would likely not be.*
>
> We have revised Sec. 2 expanding the explanations and introducing additional pointers to previous works. We are happy to further improve our presentation if the reviewer has more specific suggestions in mind.
>
> Due to space limitations, we address your question in a following comment.
>
> We hope our response addresses your concerns and welcome further suggestions to improve the clarity of our presentation. If you find our response convincing/useful, please consider increasing the score.

---

> > ### Author Response · Authors · 2024-11-19
> > **Answer to Reviewer vgxp's question**
> >
> > > *It would be nice to see if some of these results transfer to harder continual learning datasets. Specifically, these results are all done on two class classification tasks. Do they hold with more classes? Also do they hold for non MLP networks? Do they hold for datasets such as split Imagenet or split CIFAR? Do they hold across more than 2 tasks?*
> >
> > Our theoretical framework is directly applicable to multi-output architectures, including classification with more than two classes, as long as the number of classes remains finite while the input dimension goes to infinity. For instance, [Cornacchia, et al. Machine Learning: Science and Technology 4 (1), 015019] have studied Bayes learning of multi-class classification in a teacher-student model, and our theory directly applies to this case.
> >
> > Regarding multiple tasks, we have now added a paragraph about the case T=3 in Appendix C.1, our results are shown in the additional Fig. 10. Also in this case, we find that the optimal strategy outperforms the heuristic benchmarks. The structure of the optimal strategies for T=3 is more complicated, but remarkably it still shares some similarities with the pseudo-optimal strategy described in the main text. Specifically, task 1 is replayed only when its loss is comparable to the losses on the other two tasks. Notably, as the number of tasks grows, the number of possible heuristic strategies expands significantly (we considered 6 different heuristics in Fig. 10), making it difficult to identify effective solutions through intuition alone. This highlights the importance of a framework for systematically determining the optimal strategy.
> >
> > Regarding non MLP architectures, we expect our framework to be directly applicable to any architecture whose “forward” training dynamics can be tracked analytically. For instance, interesting cases that could be directly studied with our framework are narrow autoencoders and single attention layers with low-rank query-key matrix, that can be cast in the framework under consideration [H. Cui, arXiv:2409.13904, 2024].
> >
> > We have run additional experiments on CIFAR10 confirming our observations on pseudo-optimal strategy striking the balance between no-replay and interleaved. The results are presented in Appendix C.2.

---

> > > ### Comment · Reviewer_vgxp · 2024-11-25
> > >
> > > Overall I am satisfied with the changes and am raising my score to recommend acceptance.

---

### Author Response · Authors · 2024-11-19
**General response to reviewer feedback**

We thank the reviewers for their valuable time and constructive comments.

All reviewers recognize the value of our work, highlighting that our theoretical framework is “interesting” (vgxp, PcHB, 4ows), “original and refreshing” (pAPx), “useful” (vgxp), and “insightful” (4ows). Additionally, all reviewers acknowledge that our theoretical results are well-supported and thoroughly validated by numerical simulations.

In response to the reviewers' suggestions for improving clarity and expanding experiments, we have made the following updates:

- Enhanced the clarity of our technical presentation, particularly in Sec. 2, and better explained our experimental findings in Sec. 3.2.
- Conducted additional simulations on learning rate annealing, benchmarking exponential and power-law schedules (Appendix C.1).
- Extended our analysis to a continual learning problem with T=3 tasks, comparing the theory-derived optimal schedule with multiple benchmarks (Appendix C.1).
- Tested the pseudo-optimal strategy with new experiments on CIFAR10 (Appendix C.2).

Detailed responses to specific reviewer comments will be provided in individual replies.

In summary, our work offers a novel and promising theoretical framework with potential practical implications, validated by extensive simulations and interpretable results. If your concerns have been addressed, we kindly ask you to consider increasing your score.

---

### Author Response · Authors · 2024-11-24
**Discussion period ending soon**

Dear Area Chair and Reviewers,

We have submitted detailed responses to all the reviewers' concerns, along with a general response summarizing the changes made. With only one day remaining in the discussion period, we kindly ask for confirmation on whether our replies have addressed the reviewers' concerns, and, if so, to consider adjusting your score accordingly.

Your feedback is crucial to improving the quality of the paper, and we would greatly appreciate your engagement before the deadline.

Thank you for your time and consideration.

Best regards,
The Authors

---

### Author Response · Authors · 2024-11-28

Dear Reviewers and AC,

We believe we have addressed the concerns raised in your reviews in the updated version of the paper. While we can no longer update the submission, we are happy to continue engaging and answer any questions that may arise during the reviewers-AC discussion phase.

Kind regards,
The Authors

---

### Author Response · Authors · 2024-12-01
**Follow-up on updated responses before discussion ends**

Dear Area Chair and Reviewers,

Thank you for your work. We believe that our revised version reflects your recommendations. In particular: We clarified the technical presentation, expanded on experimental findings, and conducted additional analyses and experiments based on your suggestions.

While Reviewers pAPx and vgxp don't seem to need further clarifications, Reviewers 4ows and PcHB expressed some concerns that we addressed in our revision.

In particular, we addressed Reviewer 4ows's concerns about the generality of our results by adding a new experiment on CIFAR10, as requested. For Reviewer PcHB, who raised questions about the problem setup and theoretical framework, we clarified that our context is well-established in the field and highlighted the originality of our contribution.

We believe our responses address the reviewers' concerns, but we are happy to discuss further if needed. If our revisions address your concerns, we kindly ask you to reflect this in your evaluation before the discussion phase ends tomorrow.

We greatly appreciate your time and insights.

Best regards,

The Authors

---

### Meta-Review · Area_Chair_dCVM · 2024-12-21

**Metareview:**

This paper theoretically approaches continual learning using ODEs (for learning curves of online SGD), applying it to teacher-student continual learning methods, which is novel. They derive an optimal learning rate schedule and study the influence of task similarity on forgetting, which is useful / significant in continual learning. There are also experiments are on Fashion MNIST (and after rebuttal, CIFAR10), which is nice to see and improves the impact and reach of the paper. Overall, the novelty and significance are key strengths of the paper, and therefore I recommend accepting (with spotlight). I think the community will benefit from reading and discussing this paper and its theory techniques for continual learning.

**Additional Comments On Reviewer Discussion:**

One limitation is the lack of learning rate experiments (Reviewer vgxp, pAPx, 4ows), but authors ran more experiments in the rebuttal (including CIFAR10 -- I would encourage the authors to consider moving a figure from Appendix C2 into the main paper). I do think that careful experiments deviating from the theory (not necessarily larger-scale experiments) would improve the paper further, as suggested by Reviewer 4ows.

Reviewer PcHB, after a detailed discussion, thinks that the problem of optimal sampling is a limitation. I do not disagree, but think that this paper makes an important step in this direction for the community.

Otherwise, many points raised by reviewers were effectively addressed by authors, such as regarding clarity and extending to 3 tasks.

---

### Decision · Program_Chairs · 2025-01-22

Accept (Poster)